# A Unified Framework for Soft Threshold Pruning

**Yanqi Chen**[1,3], **Zhengyu Ma**[†3], **Wei Fang**[1,3], **Xiawu Zheng**[3], **Zhaofei Yu**[1,2,3], **Yonghong Tian**[†1,3]

[1]National Engineering Research Center of Visual Technology, School of Computer Science, Peking University; [2]Institute for Artificial Intelligence, Peking University; [3]Peng Cheng Laboratory
`yhtian@pku.edu.cn, mazhy@pcl.ac.cn`; ([†]Corresponding author)

## Abstract

Soft threshold pruning is among the cutting-edge pruning methods with state-of-the-art performance[1]. However, previous methods either perform aimless searching on the threshold scheduler or simply set the threshold trainable, lacking theoretical explanation from a unified perspective. In this work, we reformulate soft threshold pruning as an implicit optimization problem solved using the *Iterative Shrinkage-Thresholding* Algorithm (ISTA), a classic method from the fields of sparse recovery and compressed sensing. Under this theoretical framework, all threshold tuning strategies proposed in previous studies of soft threshold pruning are concluded as different styles of tuning $L_1$-regularization term. We further derive an optimal threshold scheduler through an in-depth study of threshold scheduling based on our framework. This scheduler keeps $L_1$-regularization coefficient stable, implying a time-invariant objective function from the perspective of optimization. In principle, the derived pruning algorithm could sparsify any mathematical model trained via SGD. We conduct extensive experiments and verify its state-of-the-art performance on both Artificial Neural Networks (ResNet-50 and MobileNet-V1) and Spiking Neural Networks (SEW ResNet-18) on ImageNet datasets. On the basis of this framework, we derive a family of pruning methods, including sparsify-during-training, early pruning, and pruning at initialization. The code is available at `https://github.com/Yanqi-Chen/LATS`.

## 1 Introduction

Pruning has been a thriving area of network compression. Since the day deep neural networks stretch their tentacles to every corner of machine learning applications, the demand for shrinking the size of network parameters has never stopped growing. Fewer parameters usually imply less computing burden on resource-constrained hardware such as embedded devices or neuromorphic chips. Some pioneering studies have revealed considerable redundancies in both Artificial Neural Networks (ANNs) (Han et al., 2015; 2016; Wen et al., 2016; Liu et al., 2017) and Spiking Neural Networks (SNNs) (Qi et al., 2018; Chen et al., 2021; Yin et al., 2021; Deng et al., 2021; Kundu et al., 2021; Kim et al., 2022b).

In essence, pruning can be formulated as an optimization problem under constraint on $L_0$ norm, the number of nonzero components in network parameters. Assuming $\mathcal{L}$ is the loss function of vectorized network weight $\boldsymbol{w}$, we expect lower $L_0$ norm $\|\boldsymbol{w}\|_0$ along with lower loss $\mathcal{L}(\boldsymbol{w})$. Despite different formulations like hard constraints

$$\min_{\mathcal{L}(\boldsymbol{w}) \leq c} \|\boldsymbol{w}\|_0; \tag{1}$$

$$\min_{\|\boldsymbol{w}\|_0 \leq K} \mathcal{L}(\boldsymbol{w}); \tag{2}$$

or soft constraints (penalized)

$$\min_{\boldsymbol{w}} \{\mathcal{L}(\boldsymbol{w}) + \mu\|\boldsymbol{w}\|_0\}, \tag{3}$$

---

[1]For example, STR (Kusupati et al., 2020) is the first to achieve >50% Top-1 accuracy of ImageNet on ResNet-50 under >99% sparsity. STDS (Chen et al., 2022) is the first pruning algorithm achieving acceptable performance degradation ($\sim$ 3% under 88.8% sparsity) for spiking neural networks with 18+ layers.

all these forms are without exception NP-Hard (Natarajan, 1995; Davis et al., 1997; Nguyen et al., 2019). Relaxing $L_0$ norm to $L_p(0 < p < 1)$ norm will not make it more tractable for it is still strongly NP-Hard (Ge et al., 2011). Nowadays, research on pruning and sparse optimization is mainly focused on the $L_1$-regularized problem, the tightest convex relaxation of $L_0$ norm, which dates back to a series of groundbreaking studies on compressed sensing (Donoho, 2006; Candès et al., 2006). These researches technically allows us to solve $L_1$-regularized problem as an alternative or, sometimes even an equivalent option (Candès, 2008) to confront $L_0$ norm constraint. A variety of modern methods such as magnitude-based pruning are still firmly rooted in solving the $L_1$ regularized optimization problem. Be that as it may, $L_1$ regularization is mostly employed for shrinking the magnitude of weight before the hard thresholding step, which has started to be replaced by other sorts of novel regularization (Zhuang et al., 2020).

In the past few years, a new range of pruning methods based on soft threshold reparameterization of weights has been developing gradually. The term "reparameterization" here refers to a specific mapping to network weights $w$ from a latent space of hidden parameters $\theta$. *The "geometry" of latent space could be designed for guiding actual weights $w$ towards sparsity*. In soft threshold pruning, the mapping is an element-wise soft threshold function with time-variant threshold. Among these studies, two representative ones are **S**oft **T**hreshold weight **R**eparameterization (**STR**) (Kusupati et al., 2020) and **S**tate **T**ransition of **D**endritic **S**pines (**STDS**) (Chen et al., 2022). They both achieve the best performance of that time. STDS further demonstrates the analogy between soft threshold mapping and a structure in biological neural systems, *i.e.*, dendritic filopodia and mature dendritic spines. However, few researchers notice that soft threshold mapping also appear as the shrinkage operator in the solution of LASSO (Tibshirani, 1996) when the design matrix is orthonormal. The studies on LASSO further derives the *Iterative Shrinkage-Thresholding* Algorithm (ISTA) (Daubechies et al., 2004; Elad, 2006), which used to be popularized in sparse recovery and compressed sensing. ISTA has many variants (Bioucas-Dias & Figueiredo, 2007; Beck & Teboulle, 2009b; Bayram & Selesnick, 2010) and has long been certified as an effective sparsification methods in all sorts of fields like deep learning (He et al., 2017; Zhang et al., 2018; Bai et al., 2020), computer vision (Beck & Teboulle, 2009a; Dong et al., 2013), medical imageology (Lustig et al., 2007; Otazo et al., 2015) and geophysics (Herrmann & Hennenfent, 2008). Despite an abecedarian analysis on the similarity between STDS and ISTA, many issues remains to be addressed, such as 1) the exact equivalence between ISTA and the growing threshold in soft threshold pruning, 2) the necessity of setting threshold trainable in STR, and 3) the way to improve existing methods without exhaustively trying different tricks for scheduling threshold.

In this work, we proposed a theoretical framework serving as a bridge between the underlying $L_1$-regularized optimization problem and threshold scheduling. The bridge is built upon the key finding that ***soft threshold pruning is an implicit ISTA for nonzero weights***. Specifically, we prove that the $L_1$ coefficient in the underlying optimization problem is determined by both threshold and learning rate. In this way, any threshold tuning strategy can now be interpreted as a scheme for tuning $L_1$ penalty. We find that a time-invariant $L_1$ coefficient lead to performance towering over previous pruning studies. Moreover, we bring a strategy of tuning $L_1$ penalty called continuation strategy (Xiao & Zhang, 2012), which was once all the rage in the field of sparse optimization, to the field of pruning. It derives broad categories of algorithms covering several tracks in the present taxonomy of pruning. In brief, our contributions are summarized as follows:

- **Theoretical cornerstone of threshold tuning strategy.** To the best of our knowledge, this is the first work interpreting increasing threshold as an ever-changing regularized term. Under theoretical analysis, we present a unified framework for the local equivalence of ISTA and soft threshold pruning. It enables us to make a comprehensive study on threshold tuning using the classic method in sparse optimization.

- **Learning rate adapted threshold scheduler.** Through our proposed framework, we reveal the strong relation between the learning rate scheduler and the threshold scheduler. Then we show that an time-invariant $L_1$ coefficient requires the changing of threshold being proportional to the learning rate. The *Learning rate Adapted Threshold Scheduler* (LATS) built upon $L_1$ coefficient achieves a state-of-the-art performance-sparsity tradeoff on both deep ANNs and SNNs.

- **Sibling schedulers cover multiple tracks of pruning.** We propose an early pruning algorithm by translating the homotopy continuation algorithm into a pruning algorithm with

our framework. It achieves indistinguishable performance to LATS as a conventional early pruning method. Moreover, the algorithm in the pruning-at-initialization setting erases some subsequent layers in ResNet and maintains the identity mapping, shrinking deep ResNet to a shallow one.

## 2 RELATED WORKS

There has been a deluge of pruning algorithms emerged since the term "deep compression" was invented. These various studies emphasize different points like granularity (structured or unstructured) and stage of pruning (at initialization, during training, post training). The difference in granularity is similar to that between LASSO and group LASSO. Empirically, unstructured pruning tends to reach higher sparsity under the same accuracy degradation. For the pruning phase, sparsify during training commonly lead to higher accuracy than early phase one, *e.g.*, pruning at initialization. Moreover, pruning during training is cheaper than post-training pruning when the overhead of dense training is considered. Some most relevant works are introduced as follows.

**Sparsify during training.** The terminology *sparsify-during-training* (also called pruning-while-learning, pruning-during-training) is mentioned in Hoefler et al. (2021), which refers to pruning and training networks simultaneously including those iteratively pruned networks. Recent works in this area includes STR (Kusupati et al., 2020), Top-KAST (Jayakumar et al., 2020), CS (Continuous Sparsification) (Savarese et al., 2020), RigL (Evci et al., 2020), WoodFisher (Singh & Alistarh, 2020), PSGD (Kim et al., 2020), GraNet (Liu et al., 2021a), Powerprop (Schwarz et al., 2021), ProbMask (Zhou et al., 2021), GPO (Wang et al., 2022), OptG (Zhang et al., 2022) and STDS (Chen et al., 2022). Many of these works are based on the reparameterization of weights using either binary mask or element-wise nonlinear mapping. The former choose to confront $L_0$ constraint directly while the latter are committed to adjusting the landscape of loss function around zero. Our method is based on soft threshold reparameterization, which is piecewise linear and has an intrinsic connection to the ISTA with $L_1$ regularization.

**Early pruning.** We refer to a variant of sparsify-during-training as early pruning here, which only exerts pruning to network in the early stage of training. It includes pruning at initialization, *e.g.*, GraSP (Wang et al., 2020), SynFlow (Tanaka et al., 2020), SBP-SR (Hayou et al., 2021), ProsPr (Alizadeh et al., 2022), and the conventional early pruning methods which stop pruning after several epochs of training (You et al., 2020; Liu et al., 2021b; Rachwan et al., 2022). Most of these works are inspired by the discovery of Lottery Ticket Hypothesis (LTH) (Frankle & Carbin, 2019) or SNIP (Lee et al., 2019), if not both. LTH suggests we can find a sparse subnetwork with comparable performance to the dense network after iterative retraining, while SNIP manages to train from an initial sparse network. A wide array of criteria like connectivity sensitivity are taken for finding such sparse networks in the early stage with promising accuracy after training.

## 3 PRELIMINARIES

**Notation.** We use $|\boldsymbol{x}|$ to denote the element-wise absolute value of $\boldsymbol{x}$, and $\|\boldsymbol{x}\|_p$ denotes the $p$-norm of $\boldsymbol{x}$. $\boldsymbol{x} \odot \boldsymbol{y}$ denotes the element-wise product of $\boldsymbol{x}$ and $\boldsymbol{y}$. If not otherwise specified, the superscript within parenthesis $\boldsymbol{x}^{(i)}$ denotes $\boldsymbol{x}$ at $i$-th iteration of gradient descent. The element-wise sigmoid function is denoted by $\sigma(\boldsymbol{x})_i := 1/(1 + e^{-x_i})$. The *soft threshold mapping* is also an element-wise mapping defined by $\mathcal{S}_d(\boldsymbol{x})_i := \text{sign}(x_i) \cdot \max\{|x_i| - d, 0\}$ with scalar threshold $d$.

### 3.1 SOFT THRESHOLD PRUNING

Basically, soft threshold pruning will iteratively execute following three core steps:

(i) **Mapping hidden weight to actual weight** $\boldsymbol{w}$ through the *soft threshold mapping* $\boldsymbol{w}^{(t)} \leftarrow \mathcal{S}_d(\boldsymbol{\theta}^{(t)})$ during training, where $\boldsymbol{\theta}$ is a trainable hidden weight with the same shape as $\boldsymbol{w}$.

(ii) **Training hidden weight** $\boldsymbol{\theta}$ through backpropagation in latent space

(iii) **Growing threshold** $d$ pushes the term $\max\{|\theta_i| - d, 0\}$ in $\mathcal{S}_d(\boldsymbol{\theta})$ towards zero and thereby enforces sparsity for $\boldsymbol{w}$.

**Algorithm 1** The general form of soft threshold pruning algorithm coupled with vanilla SGD (STR and STDS for instance).

---

**Input:** initialized network parameters $\boldsymbol{w}^{(0)}$, threshold scheduler function $g(\cdot)$, initial threshold $d^{(0)}$, final threshold $D$, initial learnable parameter $s_{\text{init}}$, loss function $\mathcal{L}(\boldsymbol{w})$, the number of training iterations $T$, $L_2$ penalty $\lambda$.
**Output:** trained sparse parameters $\boldsymbol{w}^{(T)}$

1:   $\boldsymbol{s}^{(0)} \leftarrow s_{\text{init}}, \boldsymbol{d}^{(0)} \leftarrow \sigma(\boldsymbol{s}^{(0)}), \boldsymbol{\theta}^{(0)} \leftarrow \boldsymbol{w}^{(0)}$        ▷ Initialization of weight and threshold
2: **for** $t = 0, 1, \ldots, T-1$ **do**
3:      $\Delta\boldsymbol{\theta}^{(t)} \leftarrow \nabla_{\boldsymbol{w}}(\mathcal{L}(\boldsymbol{w}^{(t)})) \odot \mathbf{1}_{|\boldsymbol{\theta}| > \boldsymbol{d}^{(t)}}$        ▷ Computing gradient with respect to hidden weight
4:      $\boldsymbol{\theta}^{(t+1)} \leftarrow \boldsymbol{\theta}^{(t)} - \eta^{(t)}(\Delta\boldsymbol{\theta}^{(t)} + \lambda\boldsymbol{\theta}^{(t)})$        ▷ Update hidden weight. $\eta^{(t)}$ is learning rate
5:      $\boldsymbol{s}^{(t+1)} \leftarrow \boldsymbol{s}^{(t)} - \eta^{(t)}(\nabla_{\boldsymbol{s}}(\mathcal{L}(\boldsymbol{w}; \boldsymbol{s}^{(t)})) + \lambda\boldsymbol{s}^{(t)})$        ▷ Threshold training in STR
6:      $\boldsymbol{d}^{(t+1)} \leftarrow \sigma(\boldsymbol{s}^{(t+1)}) g((t+1)/T) \cdot D$        ▷ Update threshold
7:      $\boldsymbol{w}^{(t+1)} \leftarrow \mathcal{S}_{\boldsymbol{d}^{(t+1)}}(\boldsymbol{\theta}^{(t+1)})$        ▷ Update the actual weight
8: **end for**
9: **return** $\boldsymbol{w}^{(T)}$

---

We provide a general form of soft threshold pruning as Algorithm 1, which is a prototype of both STR and STDS. GPO change the mapping in line 7 to a convex combination of soft threshold and identity mapping, leading to a slight difference. However, GPO only obtain marginal performance improvement with respect to STR, and will soon degenerate to STR as discussed in Appendix C. Therefore, the following of this paper are focused on STR and STDS. The differences between them are concluded from two aspects.

**Gradient computing.** With the reparameterization mapping $\boldsymbol{w}^{(t)} \leftarrow \mathcal{S}_d(\boldsymbol{\theta}^{(t)})$, the forward step includes mapping via $\boldsymbol{\theta} \to \boldsymbol{w}$ and evaluating loss with $\boldsymbol{w}$ as the actual network weight. Hence, gradient is backpropagated via path $\mathcal{L} \to \boldsymbol{w} \to \boldsymbol{\theta}$. The learning rule is thus used for updating $\boldsymbol{\theta}$ instead of $\boldsymbol{w}$. Note that $\mathcal{S}_d$ is non-differentiable at $\pm d$ and has zero gradient in interval $(-d, d)$. STR takes advantage of the subgradient at $\pm d$ and leaves zero gradients as it is. STDS views $\mathcal{S}_d$ as an identity mapping during backward, and provides a convergence analysis by approximating $\mathcal{S}_d$ using a smooth surrogate $\hat{\mathcal{S}}_d(x) = \frac{1}{\alpha}[\zeta(\alpha(x-d)) - \zeta(-\alpha(x+d))]$, where $\zeta(x) := \log(1 + e^x)$ denotes the softplus function.

**Threshold tuning.** These studies spontaneously try manipulating threshold $d$ for different sparsity. STR assigns an independent threshold for each layer, resulting into a threshold vector $\boldsymbol{d}$. Besides, STR further parameterizes the threshold by another trainable parameter $\boldsymbol{s}$. The mapping from $\boldsymbol{s}$ to $\boldsymbol{d}$ is individually designed for CNN and RNN. Compared to STR, STDS set threshold manually by introducing the **threshold scheduler** as $d^{(t)} = g(t/T) \cdot D$, wherein **scheduler function** $g : [0, 1] \to [0, 1]$ is increasing and satisfies $g(0) = 0, g(1) = 1$, $T$ is the total training steps. The formulation is based on the idea that increasing threshold from 0 to $D$ could follow different paths. The **final threshold** $D$ is the only adjustable hyperparameter for different sparsity levels when $g$ is given. Larger $D$ always leads to higher sparsity in practice.

The above techniques are summarized in Tab. 1.

Table 1: Techniques used in previous studies.

| Method | Reparameterization mapping $\boldsymbol{w}(\boldsymbol{\theta}, d)$ | Gradient of mapping | Threshold | Note |
|---|---|---|---|---|
| STR (Kusupati et al., 2020) | $\boldsymbol{w} = \mathcal{S}_d(\boldsymbol{\theta})$ | Subgradient | $d := \begin{cases} \sigma(s), & \text{for CNN} \\ e^s, & \text{for RNN} \end{cases}$ | $s$ is layer-wise (global for STR-GS), trainable, and $L_2$ regularized |
| GPO (Wang et al., 2022) | $\boldsymbol{w} = (1-k)\mathcal{S}_d(\boldsymbol{\theta}) + k\boldsymbol{\theta}$ | Gradient | $d := \sigma(s)$ | $s, k$ are layer-wise, trainable, and $L_2$ regularized |
| STDS (Chen et al., 2022) | $\boldsymbol{w} = \mathcal{S}_d(\boldsymbol{\theta})$ | $\mathbf{1}$ (Viewed as identity) | $d^{(t)} = g(t/T) \cdot D$ | $g$ is increasing, $g(0) = 0, g(1) = 1$ |

## 3.2 ITERATIVE SHRINKAGE-THRESHOLDING ALGORITHM

ISTA is initially derived from solving linear inverse problem with regularization $\min_{\boldsymbol{x} \in \mathbb{R}^n} \{\|\boldsymbol{A}\boldsymbol{x} - \boldsymbol{b}\|_2^2 + r(\boldsymbol{x})\}$, where $\boldsymbol{A} \in \mathbb{R}^{m \times n}$ and $\boldsymbol{b} \in \mathbb{R}^m$. ISTA is later extended to general objective as $\min_{\boldsymbol{x} \in \mathbb{R}^n} \{F(\boldsymbol{x}) := f(\boldsymbol{x}) + r(\boldsymbol{x})\}$ with assumptions listed below

(i) Objective function $f : \mathbb{R}^n \to \mathbb{R}$ is continuous differentiable, and $L$-smooth, *i.e.*, $\|\nabla f(\boldsymbol{x}) - \nabla f(\boldsymbol{y})\|_2 \leq L_f \|\boldsymbol{x} - \boldsymbol{y}\|_2, \forall \boldsymbol{x}, \boldsymbol{y} \in \mathbb{R}^n$, where $L_f > 0$ is the Lipschitz constant of $\nabla f$.

(ii) Regularization function $r : \mathbb{R}^n \to \mathbb{R}$ is continuous convex and can be nonsmooth.

(iii) $F$ is bounded from below.

Leave the regularization $r(\boldsymbol{x})$ alone, applying vanilla SGD to $f$ can be viewed as iteratively calculating proximal regularization (Martinet, 1970) of the linearized $f$ at $\boldsymbol{x}$, which is suggested by the following fact: $\boldsymbol{x} - \eta \nabla f(\boldsymbol{x}) = \arg\min_{\boldsymbol{y}} \{ f(\boldsymbol{x}) + \langle \boldsymbol{y} - \boldsymbol{x}, \nabla f(\boldsymbol{x}) \rangle + \frac{1}{2\eta} \|\boldsymbol{y} - \boldsymbol{x}\|_2^2 \}$, where $\eta$ is explained as "stepsize" in optimization or "learning rate" in the context of deep learning. For a given point $\boldsymbol{x}$, $F(\boldsymbol{y})$ can be approximated by expanding $f$ to the quadratic term in a similar vein

$$\hat{F}_\eta(\boldsymbol{y}; \boldsymbol{x}) := f(\boldsymbol{x}) + \langle \boldsymbol{y} - \boldsymbol{x}, \nabla f(\boldsymbol{x}) \rangle + \frac{1}{2\eta} \|\boldsymbol{y} - \boldsymbol{x}\|_2^2 + r(\boldsymbol{y}), \tag{4}$$

The above problem admits a unique minimizer

$$\arg\min_{\boldsymbol{y}} \hat{F}_\eta(\boldsymbol{y}; \boldsymbol{x}) = \arg\min_{\boldsymbol{y}} \left\{ \frac{1}{2\eta} \|\boldsymbol{y} - (\boldsymbol{x} - \eta \nabla f(\boldsymbol{x}))\|_2^2 + r(\boldsymbol{y}) \right\}. \tag{5}$$

Note that $\eta = \frac{1}{L_f}$ gives a upperbound of $f(\boldsymbol{y})$, which is obtained by the descent lemma $f(\boldsymbol{y}) \leq f(\boldsymbol{x}) + \langle \boldsymbol{y} - \boldsymbol{x}, \nabla f(\boldsymbol{x}) \rangle + \frac{L_f}{2} \|\boldsymbol{y} - \boldsymbol{x}\|_2^2$ (Beck, 2017). It implies with proper choice of $\eta$, we are virtually optimizing the upperbound of $F(\boldsymbol{y})$ using minimizer in Eq. 5. The general form of ISTA iteratively solves Eq. 5 as $\boldsymbol{x}^{(t+1)} = \arg\min_{\boldsymbol{x}} \hat{F}_\eta(\boldsymbol{x}; \boldsymbol{x}^{(t)})$, which is also known as proximal gradient methods (Combettes & Wajs, 2005). The detailed convergence analysis is presented in vast optimization literature like FISTA (Beck & Teboulle, 2009b) and GIST (Gong et al., 2013). For sparsity, we are interested in $L_1$ regularization term $r(\boldsymbol{x}) = \mu \|\boldsymbol{x}\|_1, \mu > 0$. Since $L_1$ norm is separable, we have a closed-form solution with element-wise soft threshold operation as

$$\boldsymbol{x}^{(t+1)} = \mathcal{S}_{\mu\eta}(\boldsymbol{x}^{(t)} - \eta \nabla f(\boldsymbol{x}^{(t)})). \tag{6}$$

Eq. 6 gives the ISTA update rule under $L_1$ regularization.

## 4 A FRAMEWORK FOR SOFT THRESHOLD PRUNING

In this part, we will formulate the growing threshold under soft threshold reparameterization as an implicit ISTA. For simplicity, we assume the threshold is global across all parameters, which is consistent with the setting of STDS and STR-GS, *i.e.*, the global threshold version of STR. Hence, we use scalar $d$ rather than vector $\boldsymbol{d}$ in Algorithm 1 to denote the global threshold.

To begin with, we investigate the update rule of nonzero components in actual weight

$$\boldsymbol{\theta}^{(t+1)} \leftarrow \boldsymbol{\theta}^{(t)} - \eta^{(t)} \nabla_{\boldsymbol{w}}(\mathcal{L}(\boldsymbol{w}^{(t)})) \odot \nabla_{\boldsymbol{\theta}}(\boldsymbol{w}(\boldsymbol{\theta}^{(t)}, d^{(t)})), \qquad \text{Line 4 in Algorithm 1,} \tag{7}$$

$$\boldsymbol{w}^{(t+1)} \leftarrow \mathcal{S}_{d^{(t+1)}}(\boldsymbol{\theta}^{(t+1)}), \qquad \text{Line 7 in Algorithm 1.} \tag{8}$$

Assuming the sign of weight remains unchanged after an update, which happens when the gradient has the opposite sign of weight or the gradient magnitude is sufficiently small, we have the following Lemma:

**Lemma 1** (Local update rule). *The update rule in Eq. 7 and Eq. 8 implies the following update of a nonzero component $w(\theta, d)$ in actual weight $\boldsymbol{w}$*

$$w^{(t+1)} = \mathcal{S}_{d^{(t+1)} - d^{(t)}}(w^{(t)} - \eta^{(t)} \nabla_w(\mathcal{L}(w^{(t)}))). \tag{9}$$

The formal version and proof of Lemma 1 is given in Appendix A. Note that the form in Eq. 9 is equivalent to the ISTA update rule with threshold equal to forward finite difference $d^{(t+1)} - d^{(t)}$. Recall Eq. 6, the corresponding optimization problem can be deduced as Theorem 1:

**Theorem 1.** *Let $\mathcal{L}(\boldsymbol{w})$ be the loss function depending on the network weight $\boldsymbol{w}$, which is further reparamterized by hidden weight $\boldsymbol{\theta}$ and threshold $d$ as $\boldsymbol{w}(\boldsymbol{\theta}, d) = \mathcal{S}_d(\boldsymbol{\theta})$. When applying vanilla*

*SGD to reparameterized nonzero weight, the update rule is locally equivalent to solving the following problem using ISTA with penalty term*

$$\min_{\boldsymbol{w}} \left\{ F(\boldsymbol{w}) := \mathcal{L}(\boldsymbol{w}) + \frac{d^{(t+1)} - d^{(t)}}{\eta^{(t)}} \|\boldsymbol{w}\|_1 \right\}, \tag{10}$$

*where $\eta^{(t)}, d^{(t)}$ denotes the learning rate and threshold at the t-th iteration, respectively. The magnitude of $L_1$-penalty is the quotient of forward finite difference of threshold scheduler divided by learning rate.*

Theorem 1 serves as the glue holding the learning rate scheduler $\eta^{(t)}$, threshold $d^{(t)}$ and penalty $\mu^{(t)}$ together. It also to some extent justify the increasing threshold for a positive penalty term. Under this framework, we further explain in Appendix B that the original STDS uses an improper threshold scheduler. Moreover, the validity of training threshold is discussed in Appendix C.

## 5 FINDING OPTIMAL THRESHOLD SCHEDULER

In this part, we are devoted to evaluating some threshold schedulers based on our theoretical framework and literature on sparse optimization. With the framework, any past strategy of tuning $L_1$ penalty can be converted to a feasible pruning algorithm today.

### 5.1 LEARNING RATE ADAPTED THRESHOLD SCHEDULER

If the optimization problem in Theorem 1 is fixed, or in other words, the $L_1$ penalty is invariant $\mu^{(t)} \equiv \mu$ during learning, we can deduce a unique threshold scheduler **LATS**. In LATS, *the change on threshold $d^{(t+1)} - d^{(t)}$ must be proportional to the learning rate $\eta^{(t)}$ during training.*

**Corollary 1** (**L**earning rate **A**dapted **T**hreshold **S**cheduler). *For fixed $L_1$ regularized problem*

$$\min_{\boldsymbol{w}} \left\{ F(\boldsymbol{w}) := \mathcal{L}(\boldsymbol{w}) + \mu\|\boldsymbol{w}\|_1 \right\}, \tag{11}$$

*where $\mu > 0$ is the time-independent $L_1$ penalty coefficient, the threshold scheduler is governed by the learning rate scheduler as $d^{(t)} = d^{(0)} + \mu \sum\limits_{i=0}^{t-1} \eta^{(i)}$*

For the most frequently used testbed, ResNet-50 (He et al., 2016) on ImageNet dataset (Deng et al., 2009), researchers usually use the cosine annealing learning rate scheduler (Loshchilov & Hutter, 2017) with $\eta_{\min} = 0$. Assuming the initial threshold is zero, simple algebra shows the definition of corresponding LATS with a slight abuse of notation

$$d^{(n,b)} = \mu\eta_{\max} \left[ \frac{B}{4} \left( 2n + 1 + \frac{\sin\left(\frac{2n-1}{2N}\pi\right)}{\sin\frac{\pi}{2N}} \right) + \frac{b}{2} \left( 1 + \cos\frac{n\pi}{N} \right) \right], \tag{12}$$

where the threshold $d^{(n,b)}$ depends on epoch id $n$ and batch id $b$. Here $n = 0, 1, \ldots, N-1$ denotes the current id of training epoch, and $b = 1, 2, \ldots, B$ denotes the batch id in $n$-th epoch. We elaborate the detailed derivation of Eq. 12 in Appendix D.

### 5.2 SIMPLIFIED THRESHOLD SCHEDULER

Computation such as Eq. 12 is intricate for implementation. In effect, painstakingly coding LATS according to given learning rate scheduler is not inevitable. To ease the computing burden, we turn to replacing sum of learning rate with integration of learning rate function.

To be specific, assuming the learning rate scheduler can be expressed by $\eta^{(n,b)} = h(n/N)$, the simplified threshold scheduler is defined by

$$d^{(t)} = d^{(N-1,B)} \cdot \frac{\int_0^{\frac{t}{T}} h(x)\mathrm{d}x}{\int_0^1 h(x)\mathrm{d}x}. \tag{13}$$

The simplification is loyal to the idea that the value of Riemann integral could be approximated by rectangle method. Eq. 13 can be interpreted with scheduler form in STDS as $d^{(t)} = g(t/T) \cdot D$

with scheduler function $g(x) = \int_0^{\frac{t}{T}} h(x)\mathrm{d}x / \int_0^1 h(x)\mathrm{d}x$ and final threshold $D = d^{(N-1,B)}$. The detailed derivation of Eq. 13 is given in Appendix D.

Now we have the **Simplified LATS** (**S-LATS** for short) for the cosine annealing learning rate scheduler $h(x) = \frac{\eta_{\max}}{2}(1 + \cos(\pi x))$

$$d^{(t)} = \frac{\mu\eta_{\max}T}{2} \cdot \frac{\int_0^{\frac{t}{T}} \frac{1}{2}(1 + \cos\pi x)\mathrm{d}x}{\int_0^1 \frac{1}{2}(1 + \cos\pi x)\mathrm{d}x} = \frac{\mu\eta_{\max}T}{2}\left[\frac{1}{\pi}\sin(\frac{t\pi}{T}) + \frac{t}{T}\right]. \tag{14}$$

The final threshold is $D = \mu\eta_{\max}T/2$, which satisfies $D \propto \mu$. Tuning $D$ is thus akin to changing the magnitude of penalty. In the following discussion, we employ the final threshold $D$ instead of $d^{(N-1,B)}$ to lighten the notation. The threshold schedulers in the rest of this work will thus be expressed in a unified form of $d^{(t)} = g(t/T) \cdot D$. We evaluate LATS and S-LATS under identical

Table 2: Comparison of LATS and S-LATS when applied to ResNet-50 on ImageNet dataset.

| Final threshold $D$ | STDS + LATS | | STDS + S-LATS | |
|---|---|---|---|---|
| | Sparsity (%) | Top-1 Acc. (%) | Sparsity (%) | Top-1 Acc. (%) |
| 0.1 | 79.97 | 76.53 | 79.95 | 76.75 |
| 0.5 | 95.54 | 73.12 | 95.53 | 73.03 |
| 1.0 | 97.43 | 69.56 | 97.43 | 69.64 |
| 5.0 | 99.27 | 53.80 | 99.28 | 53.64 |

final thresholds $D = 0.1, 0.5, 1.0, 5.0$, which can be gleaned from Tab. 2. The results show LATS and S-LATS are indistinguishable from accuracy and sparsity. Thus, we turn to a simplified threshold scheduler in the following discussion.

### 5.3 Continuation strategy

Also known as "warm starting", *continuation strategy* is designated for accelerating convergence. Similar to annealing of learning rate, continuation refers to gradually reducing $L_1$ penalty during learning. It is also explained in Hale et al. (2008) as an analogue to the homotopy algorithms in statistics. Continuation method used to serve as a common trick in abundant classic literature concerning sparse optimization including GPSR (Figueiredo et al., 2007), fixed point continuation method (Hale et al., 2008), SpaRSA (Wright et al., 2009) and NESTA (Becker et al., 2011).

#### 5.3.1 PGH scheduler

In the series works of *proximal gradient homotopy* (PGH) (Xiao & Zhang, 2012; 2013; Lin & Xiao, 2014), the researchers provide proof of geometric convergence rate when inducing exponentially decaying $L_1$ coefficient $\mu^{(t)} = \beta^{\frac{t}{T}}$, where $0 < \beta < 1$ is a constant. Considering our formulation of soft threshold pruning in Theorem 1, PGH can be translated into the **PGH scheduler** (simplified using Eq.13), which can be written as

$$d^{(t)} = g_{\mathrm{PGH}}(t/T) \cdot D := \frac{\int_0^{\frac{t}{T}} \frac{1}{2}(1 + \cos\pi x)\beta^x\mathrm{d}x}{\int_0^1 \frac{1}{2}(1 + \cos\pi x)\beta^x\mathrm{d}x} \cdot D, \tag{15}$$

where the analytic form of $g_{\mathrm{PGH}}$ is shown below

$$g_{\mathrm{PGH}}(x) = \frac{\pi^2\left(\beta^x - 1\right) + \log^2(\beta)\left(\beta^x - 2\right) + \log(\beta)\beta^x\left(\log(\beta)\cos(\pi x) + \pi\sin(\pi x)\right)}{\pi^2(\beta - 1) - 2\log^2(\beta)}. \tag{16}$$

#### 5.3.2 Link to early pruning

For the PGH scheduler, we interpolate $\beta$ between 0 and 1 and get a series of different PGH schedulers. As shown in Fig. 1, the increasing in threshold slows over time, which is caused by decaying $L_1$ penalty. For $0 < \beta < 1$, if pruning is ignored when penalty is below a preset threshold, we get a family of early pruning algorithms. They will stop pruning at different stages. Recall that the regularized term is proportional to the forward finite difference and thus can be approximated by derivative, for conventional early pruning, we regard $g'_{\mathrm{PGH}}(t/T) < 0.1$ as the termination criterion of pruning.

There are two limit cases when $\beta$ approaches 0 or 1. It is obvious that $\beta \to 1$ leads to S-LATS for no decay is applied. When $\beta \to 0$, the penalty is always zero except for the beginning. In this case, PGH scheduler degenerates to a magnitude-based pruning after weight initialization followed by a normal training stage. This is also referred to as *sparse-to-sparse training* or *pruning at initialization* method.

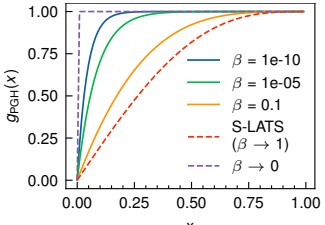

Figure 1: PGH schedulers under different $\beta$.

## 6 EXPERIMENTS

In this section, we test proposed threshold scheduler S-LATS on both deep ANNs and SNNs. The favorable performances against the previous studies are confirmed. In all experiments we switch sparsity levels by changing $D$, which is equivalent to tuning $L_1$ penalty coefficient. We also tune hyperparameter $\beta$ in PGH scheduler to maintain different phases of early pruning algorithm. Compared to the dense baseline, no tuning on other training hyperparameters is needed, which minimize the effort when applying to other networks.

### 6.1 S-LATS

S-LATS achieves state-of-the-art performances on both ANNs (ResNet-50, MobileNet-V1 (Howard et al., 2017)) and SNNs (SEW ResNet-18 (Fang et al., 2021)). The results on ResNet-like networks and MobileNet-V1 are illustrated in Fig. 2. We add Gradual Magnitude Pruning (GMP) (Zhu & Gupta, 2017) into comparison for few recent studies are conducted on MobileNet-V1. Notably, our method surges ahead of all the other baselines under <98% sparsity for pruning on ResNet-50, which is shown in Tab. 3. It should be noted that the origin STDS excludes the last FC layer from pruning in SNNs, while the results reported here are shown by rerunning it with the last layer pruned.

We also admit it fails to achieve comparable performance with a few baselines like OptG and in

Table 3: Comparison of ResNet-50 Top-1 accuracy on ImageNet in recent studies using 100 training epochs. For some studies without strict control on sparsity, the closest sparsity is reported behind the performance. Performance of S-LATS is averaged over three trials.

| Method | Batch size | Sparsity | | | | | | |
|---|---|---|---|---|---|---|---|---|
| | | 80% | 90% | 95% | 96.5% | 97.5% | 98% | 99% |
| STR | 256 | 76.19 (79.55) | 74.31 (90.23) | 70.40 (95.03) | 67.22 (96.53) | - | 61.46 (98.05) | 51.82 (98.98) |
| STR-GS | 256 | - | 74.13 (89.54) | - | - | - | 62.17 (97.91) | - |
| GraNet | 256 | 76 | 74.5 | 72.3 | 70.5 | - | - | - |
| ProbMask | 256 | - | 74.68 | 71.5 | - | - | 66.83 | 61.07 |
| OptG | 256 | - | 74.28 | 72.38 | 70.85 | - | 67.2 | **62.1** |
| WoodFisher | 256 | **76.73** | 75.26 | 72.16 | - | - | 65.47 | - |
| **S-LATS** | 256 | $76.57_{\pm0.15}$ (79.95) | $\mathbf{75.43_{\pm0.17}}$ (89.57) | $\mathbf{73.20_{\pm0.09}}$ (95.12) | $\mathbf{71.48_{\pm0.13}}$ (96.58) | $\mathbf{69.49_{\pm0.18}}$ (97.43) | $67.25_{\pm0.19}$ (98.01) | $58.39_{\pm0.25}$ (99.02) |
| **S-LATS** | 1024 | $\mathbf{76.61_{\pm0.25}}$ (79.00) | $\mathbf{75.87_{\pm0.15}}$ (90.15) | $\mathbf{74.29_{\pm0.28}}$ (95.01) | $\mathbf{72.80_{\pm0.18}}$ (96.53) | $\mathbf{70.78_{\pm0.09}}$ (97.54) | $69.15_{\pm0.13}$ (98.00) | $\mathbf{61.90_{\pm0.18}}$ (98.93) |
| RigL (ERK) | 4096 | 75.1 | 73.0 | 69.7 | 67.2 | - | - | - |
| Top-KAST (Powerprop) | 4096 | 76.24 | 75.23 | 73.25 | - | - | - | - |
| Top-KAST (Powerprop+ERK) | 4096 | 76.76 | 75.74 | - | - | - | - | - |

extreme sparsity ($\geq 99\%$), which suggests the theory might be imperfect under such conditions. An ablation study shows S-LATS outperforms the default threshold scheduler of STDS, which is shown in Appendix F.

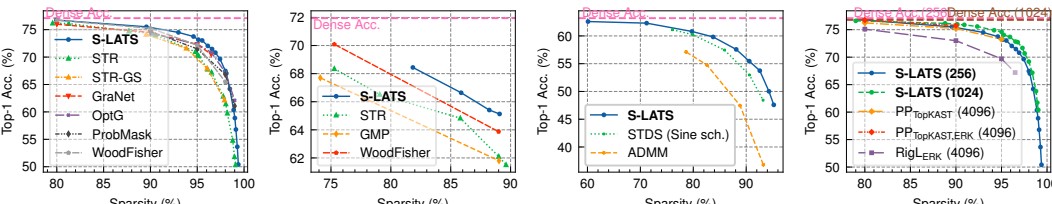

Figure 2: Performance of several SOTA pruning strategies of **ResNet-50 (Leftmost & Rightmost)**, **MobileNet-V1 (Middle left)** and **SEW ResNet-18 (Middle right)** on ImageNet. All trials uses the standard training setting (256 batch size) except the rightmost one, which uses an enlarged batch size marked in parenthesis. Detailed layerwise sparsity and accuracy are given in Appendix I.

**Pruning using large batch size.** Powerprop (Schwarz et al., 2021), adopts a batch size of 4096 for pruning of ResNet-50 and achieves fascinating performances under high sparsity. Hence, we explore the large batch size setting. Due to limited resources, we only increase it to 1024 and enlarge the learning rate correspondingly. The rightmost of Fig. 2 shows it indeed leads to higher performance, which outperforms all other SOTA studies. Astonishingly, even though the performance of the dense network slightly degrades, the accuracy of the sparse ones is improved overall. On the basis of the above finding, we believe applying an even larger batch size, like 4096 used in Powerprop, to our method may lead to top-notch performance tradeoff.

## 6.2 PGH SCHEDULER

**Conventional early pruning** Our experiments of early pruning includes $\beta = 0.1, 10^{-5}, 10^{-10}$. The corresponding ending criteria $t/T = 0.743, 0.382, 0.231$ are given by numerical solution. It indicates pruning roughly stops at the 74th, the 38th and the 23th epoch. The network using PGH scheduler with smaller $\beta$ converges faster to a sparse one, which is illustrated in the left of Fig. 3. Surprisingly, we find in the middle of Fig. 3 that for different $\beta$, the datapoint of accuracy against

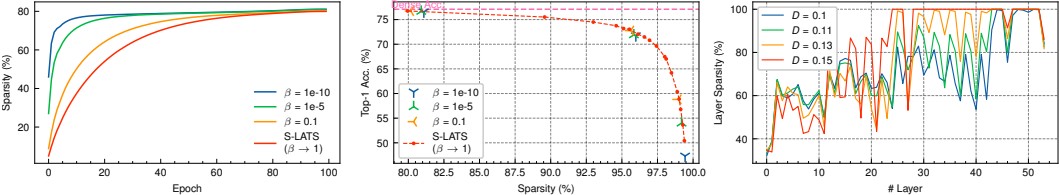

Figure 3: Overall sparsity during learning when final threshold $D = 0.1$ (**Left**). Performance under different sparsity levels (**Middle**). Layerwise sparsity of PGH scheduler under pruning at initialization setting $\beta \to 0$ (**Right**).

sparsity almost lies on the curve of S-LATS. It suggests these schedulers have practically the same performances as S-LATS, but with faster convergence to sparse networks. With the help of PGH scheduler, we are able to find sparse networks earlier with negligible performance degradation.

**Pruning at initialization** We also try $\beta \to 0$, which refers to increasing the threshold to its maximum at the first iteration. Note that our method is agnostic about the structure of network. Hence, some layers are completely pruned as shown in the right of Fig. 3, wherein three consecutive layers within a residual block tend to be pruned simultaneously. However, owing to the skip connection in ResNet, the feature can still pass through shortcuts to the final FC layer, and thus the whole networks are still normally trained. The aforementioned results are collected in Tab. 4.

Table 4: Results in pruning at initialization experiments.

| Final threshold $D$ | 0.1 | 0.11 | 0.13 | 0.15 |
|---|---|---|---|---|
| Overall sparsity (%) | 87.16 | 90.00 | 93.11 | 95.64 |
| Top-1 Acc. (%) | 74.69 | 72.89 | 68.23 | 62.22 |
| # Zeroed layers | 0 | 9 | 9 | 27 |

## 7 CONCLUSION & DISCUSSION

In this work, we present a framework interpreting increasing threshold as a constantly changing penalty term and reveal the underlying connection between soft threshold pruning and ISTA. We also derive a couple of threshold schedulers, which achieve comparable performance to current SOTA works and cover multiple tracks of pruning. It is worth noting that our method is agnostic about the object of pruning. This design endows our method with versatility while treating weight wheresoever equally, and yet becomes totally ignorant of nowadays pruning researches like sparsity budget allocation, *e.g.*, *Erdős-Rényi* (Mocanu et al., 2018) and *Erdős-Rényi-Kernel (ERK)* (Evci et al., 2020), or commonsense in this area like "Leave at least one path from input through output". We believe our method can for sure further benefit from knowledge in the prosperous field of network pruning.

ACKNOWLEDGMENTS

This work is supported by grants from the National Key R&D Program of China under Grant 2020AAA01035, the Key-Area Research Development Program of Guangdong Province (2021B0101400002), and the National Natural Science Foundation of China under contract No. 62088102, No. 62176003, No. 62006132, No. 62027804 and No. 61825101. The computing resources of Pengcheng Cloudbrain are used in this research.

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

## A PROOF OF THEOREMS AND LEMMAS

For clarity, we restate the theorem or lemma in the main text again here.

**Lemma 1** (Local update rule). *The update rule below*

$$\boldsymbol{\theta}^{(t+1)} \leftarrow \boldsymbol{\theta}^{(t)} - \eta^{(t)} \nabla_{\boldsymbol{w}}(\mathcal{L}(\boldsymbol{w}^{(t)})) \odot \nabla_{\boldsymbol{\theta}}(\boldsymbol{w}(\boldsymbol{\theta}^{(t)}, d^{(t)})) \tag{17}$$

$$\boldsymbol{w}^{(t+1)} \leftarrow \mathcal{S}_{d^{(t+1)}}(\boldsymbol{\theta}^{(t+1)}) \tag{18}$$

*imply the following update of any nonzero component $w(\theta, d)$ in actual weight $\boldsymbol{w}$*

$$w^{(t+1)} = \mathcal{S}_{d^{(t+1)} - d^{(t)}}(w^{(t)} - \eta^{(t)} \nabla_w(\mathcal{L}(w^{(t)}))). \tag{19}$$

*when $|\theta^{(t+1)}| > d^{(t)}$ and the sign condition $\text{sign}(\theta^{(t+1)}) = \text{sign}(\theta^{(t)})$ are met.*

*Proof.* For any nonzero weight $w^{(t)} \neq 0$, $\theta^{(t)} = w^{(t)} + d^{(t)} \text{sign}(w^{(t)})$, using Eq. 17 we have

$$\theta^{(t+1)} = w^{(t)} + d^{(t)} \text{sign}(w^{(t)}) - \eta^{(t)} \nabla_w(\mathcal{L}(w^{(t)})) \tag{20}$$

Let $\bar{w}^{(t+1)} := w^{(t)} - \eta^{(t)} \nabla_w(\mathcal{L}(w^{(t)}))$ be the target point of vanilla SGD without regularization. Recall Eq. 20, we have

$$
\begin{aligned}
\bar{w}^{(t+1)} &= \theta^{(t+1)} - d^{(t)} \text{sign}(w^{(t)}) \\
&= \text{sign}(\theta^{(t+1)})|\theta^{(t+1)}| - \text{sign}(\theta^{(t)})d^{(t)} \\
&= \text{sign}(\theta^{(t+1)})|\theta^{(t+1)}| - \text{sign}(\theta^{(t+1)})d^{(t)} \\
&= \text{sign}(\theta^{(t+1)})(|\theta^{(t+1)}| - d^{(t)}),
\end{aligned}
\tag{21}
$$

which has the same sign as $\theta^{(t+1)}$. Now we have $\text{sign}(w^{(t+1)}) = \text{sign}(\theta^{(t+1)}) = \text{sign}(\theta^{(t)}) = \text{sign}(w^{(t)}) = \text{sign}(\bar{w}^{(t+1)})$.

To evaluate the updated weight, by Eq. 18, Eq. 20, Eq. 21 and the definition of soft threshold mapping, we derive

$$
\begin{aligned}
w^{(t+1)} &= \text{sign}(\theta^{(t+1)}) \max\{|w^{(t)} + d^{(t)} \text{sign}(w^{(t)}) - \eta^{(t)} \nabla_w(\mathcal{L}(w^{(t)}))| - d^{(t+1)}, 0\} \\
&= \text{sign}(\theta^{(t+1)}) \max\{|\bar{w}^{(t+1)} + d^{(t)} \text{sign}(\bar{w}^{(t+1)})| - d^{(t+1)}, 0\} \\
&= \text{sign}(\theta^{(t+1)}) \max\{|\text{sign}(\bar{w}^{(t+1)})(|\bar{w}^{(t+1)}| + d^{(t)})| - d^{(t+1)}, 0\} \\
&= \text{sign}(\theta^{(t+1)}) \max\{|\bar{w}^{(t+1)}| + d^{(t)} - d^{(t+1)}, 0\} \\
&= \text{sign}(\bar{w}^{(t+1)}) \max\{|\bar{w}^{(t+1)}| - (d^{(t+1)} - d^{(t)}), 0\} \\
&= \mathcal{S}_{d^{(t+1)} - d^{(t)}}(w^{(t)} - \eta^{(t)} \nabla_w(\mathcal{L}(w^{(t)}))).
\end{aligned}
\tag{22}
$$

$\square$

## B ORIGINAL THRESHOLD SCHEDULER IN STDS

In STDS, the authors propose the Sine scheduler $d^{(t)} = \frac{1}{2}(1 + \sin(\pi(\frac{t}{T} - \frac{1}{2})))D = \frac{1}{2}(1 - \cos(\frac{t\pi}{T}))D$. With the form of simplified threshold scheduler in Eq. 13, by Theorem 1, the corresponding penalty

$\mu^{(t)}$ has the form

$$
\begin{aligned}
\mu^{(t)} &= \frac{d^{(t+1)} - d^{(t)}}{\eta^{(t)}} \\
&= D \cdot \frac{\cos(\frac{t\pi}{T}) - \cos(\frac{(t+1)\pi}{T})}{\eta_{\max}(1 + \cos\frac{t\pi}{T})} \\
&= \frac{2D\sin\frac{\pi}{T}}{\eta_{\max}} \cdot \frac{\sin(\frac{t\pi}{T} + \frac{\pi}{2T})}{1 + \cos\frac{t\pi}{T}} \\
&\approx C \cdot \frac{\sin(\frac{t\pi}{T})}{1 + \cos\frac{t\pi}{T}} \\
&= C \cdot \tan(\frac{t\pi}{2T})
\end{aligned}
\tag{23}
$$

It is a function of training progress $t/T$ with constant $C$. Investigate the function $\tan(x/2)$ on interval $(0, \pi)$, we have $\mu(x)$ is increasing from 0 to $+\infty$, which implies $\mu$ can be sufficiently large during training. The loss is thus insignificant compared to the regularization term in the last stage of training and leads to performance degradation with respect to S-LATS.

## C    DISCUSSION ABOUT TRAINING THRESHOLD

In the main text, we propose a framework explaining the style of growth in threshold as an ever-changing optimization problem. However, we only cover manually designed threshold schedulers. We make a discussion here and show training threshold is not as easy as STR or GPO did. We will show GPO and STR share the same discussion of training threshold since GPO will degenerate to STR in a few training epochs. Moreover, we suggest not simply setting threshold trainable if one really wants to investigate optimization of threshold.

### C.1    $L_2$ PENALTY DOMINATES EARLY TRAINING OF STR.

In the official codebase of STR, we notice the trainable sparse threshold is also together with weight decay ($L_2$ regularization $\lambda\|s\|_2^2$). $\lambda$ is of magnitude around $10^{-5} \sim 10^{-4}$. The initial value $s_{\text{init}}$ is usually set to negative number with large magnitude around $-10^4 \sim -10^3$ for CNN trials. It is easy to see the magnitude of $L_2$ regularized term is around $0.01 \sim 1$ in the early stage of training.

Take CNN for instance. For given loss function $\mathcal{L}$ and weight $\boldsymbol{w}_l$ of the $l$-th layer, the gradient passed to threshold $s_l$ can be estimated as

$$
\begin{aligned}
\nabla_{s_l}\mathcal{L}(\boldsymbol{w}_l(s_l, \boldsymbol{\theta}_l)) &= \left\langle \nabla_{\boldsymbol{w}_l}\mathcal{L}(\boldsymbol{w}_l), \left(\frac{\partial \boldsymbol{w}_l}{\partial s_l}\right)^\top \right\rangle \\
&= \sum_{(w,\theta) \in (\boldsymbol{w}_l, \boldsymbol{\theta}_l)} \nabla_w\mathcal{L}(\boldsymbol{w}_l) \cdot \nabla_{s_l}(\mathcal{S}_{\sigma(s_l)}(\theta)) \\
&= -\sigma(s_l)(1 - \sigma(s_l)) \sum_{\substack{(w,\theta) \in (\boldsymbol{w}_l, \boldsymbol{\theta}_l) \\ w \neq 0}} \nabla_w\mathcal{L}(\boldsymbol{w}_l) \cdot \text{sign}(\theta)
\end{aligned}
\tag{24}
$$

wherein each term of sum has magnitude less than $\frac{1}{4}\nabla_w\mathcal{L}(\boldsymbol{w}_l)$ since sigmoid function $\sigma(x)$ is bounded in $(0, 1)$. Given that the stochastic gradient noise across parameters in $\boldsymbol{w}_l$ admits to Lévy distribution (Xie et al., 2021), some negative and positive terms will balance each other, which makes us wonder whether the regularization term dominates the training of threshold. Therefore, we compare the gradient to $L_2$ penalty by tracing the magnitude ratio of $\nabla_{s_l}\mathcal{L}$ to $\lambda|s_l|$ during training of ResNet-50 on ImageNet, which is shown in Fig. 4.

It is evident that the $L_2$ penalty plays the leading role in the early stage rather than the gradient. We even observe for several final layers, the penalty always dominates the training of threshold. The update of $s$ can thereby be rewritten to $s^{(t+1)} \approx s^{(t)}(1 - \eta^{(t)}\lambda)$, where $\eta^{(t)}$ is the learning rate at the

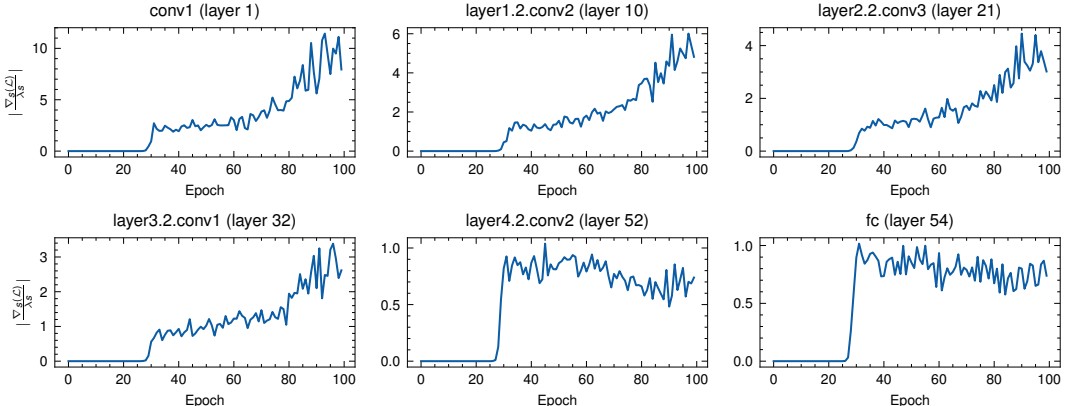

Figure 4: Magnitude ratios of gradient to $L_2$-regularization $\left|\frac{\nabla_{s_l}\mathcal{L}}{\lambda s_l}\right|$ in different layers when training using STR at 90.23% overall sparsity. Data within an epoch are averaged using geometric mean.

$t$-th iteration. Regardless of gradient, at the beginning of training, STR can be viewed as a special case of threshold scheduler as follows

$$d^{(t)} = \sigma\left(s_{\text{init}} \cdot \prod_{i=0}^{t-1}(1 - \eta^{(i)}\lambda)\right), t = 1, 2, \ldots \tag{25}$$

By Theorem 1, the corresponding penalty can be derived by

$$
\begin{aligned}
\mu^{(t)} &= \frac{d^{(t+1)} - d^{(t)}}{\eta^{(t)}} \\
&= \frac{\sigma(s^{(t+1)}) - \sigma(s^{(t)})}{\eta^{(t)}} \\
&= \frac{\sigma'(\bar{s})(s^{(t+1)} - s^{(t)})}{\eta^{(t)}} \\
&= -\frac{\sigma'(\bar{s})s^{(t)}\eta^{(t)}\lambda}{\eta^{(t)}} \\
&= -\sigma'(\bar{s})s^{(t)}\lambda
\end{aligned}
\tag{26}
$$

where $\bar{s}$ lies between $s^{(t+1)}$ and $s^{(t)}$. The existence of $\bar{s}$ is shown by Lagrange's Mean Value Theorem. It is rather difficult to analytically give the explicit expression of $\mu^{(t)}$ since it relies on the behavior of $s$, which has a dynamic decay rate $1 - \eta^{(t)}\lambda$.

Loosely speaking, we adopt the approximation $\bar{s} \approx s^{(t)}$. We still cannot show the explicit form of $\mu^{(t)}$, but now we can investigate the trend of penalty by analyzing function $\mu(x) = -\sigma'(x)x$, which is increasing in $(-\infty, \gamma)$. Here $\gamma \approx -1.5434$ is the unique negative root of $e^{-x}(x+1) - x + 1 = 0$. In the early stage, $s$ is a negative number with a much larger magnitude than $\gamma$. Based on the above, $\mu$ is increasing in the early stage.

## C.2  GPO: FALL BACK TO STR SHORTLY

It concludes STR is majorly influenced by $L_2$ decay on $s$. We will see that GPO has a similar behavior on $k$ and GPO will degenerate to STR shortly after the training begins.

In GPO, the authors introduces another trainable parameter $\beta$ in the reparameterization $\boldsymbol{w} = (1 - k)\mathcal{S}_d(\boldsymbol{\theta}) + k\boldsymbol{\theta}$ with $k = 10^{-6}|\beta|$. GPO starts from the identity mapping $\boldsymbol{w} = \boldsymbol{\theta}$ by initializing

$\beta = 10^6$. The gradient passed to $\beta_l$ in the $l$-th layer is given by

$$
\nabla_{\beta_l} \mathcal{L}(\boldsymbol{w}_l(s_l, \beta_l, \boldsymbol{\theta}_l)) = \left\langle \nabla_{\boldsymbol{w}_l} \mathcal{L}(\boldsymbol{w}_l), \left( \frac{\partial \boldsymbol{w}_l}{\partial \beta_l} \right)^\top \right\rangle
$$
$$
= 10^{-6} \operatorname{sign}(\beta_l) \cdot \sum_{(w, \theta) \in (\boldsymbol{w}_l, \boldsymbol{\theta}_l)} \nabla_w \mathcal{L}(\boldsymbol{w}_l) \cdot (\theta - \mathcal{S}_d(\theta))
$$
(27)

Notice that $\theta - \mathcal{S}_d(\theta) = \begin{cases} \theta, & |\theta| < d \\ d \operatorname{sign}(\theta), & |\theta| \geq d \end{cases}$ has magnitude not greater than $d$, which derives that the gradient $\nabla_w \mathcal{L}(\boldsymbol{w}_l)$ are added with coefficient whose magnitude is below $10^{-6}d$.

From the codebase of GPO, we confirm the weight decay on $\beta$ is $10^{-4}$. Since the initial $\beta$ is $10^6$, the $L_2$ regularized term is of magnitude around $10^{-4} \times 10^6 = 100$ at the beginning of training. Recall that $d = \sigma(s) < 1$, it is obvious that the gradient pass to $\beta$ has a much smaller magnitude than the $L_2$ regularization. For this reason, the gradient can be ignored for $k$. Furthermore, $k$ will shrink exponentially to almost zero and the mapping in GPO will fall back to STR within a few epochs, which can be seen by $\boldsymbol{w} = (1 - k)\mathcal{S}_d(\boldsymbol{\theta}) + k\boldsymbol{\theta} \approx \mathcal{S}_d(\boldsymbol{\theta})$.

### C.3 FOCUS ON THRESHOLD SCHEDULER INSTEAD OF THE FINAL THRESHOLD.

Conventional wisdom suggests that when a parameter is set trainable, it will be optimized automatically. This idea should only work for those directly determining the performance, *e.g.*, weights in a dense network. For pruning, authors of STDS find the differences in positions of performance versus sparsity curve should be ascribed to the evolving patterns of the threshold. To verify this, we replace the threshold training mechanism in STR and default scheduler in STDS by several schedulers with the same final threshold $D = 10^{-3}$ ($D = 0.5$ for STDS) shown below

- Sine: $d^{(t)} = \frac{1}{2}(1 + \sin(\pi(\frac{t}{T} - \frac{1}{2})))D$
- Linear: $d^{(t)} = \frac{t}{T}D$
- Log2: $d^{(t)} = \log_2(\frac{t}{T} + 1)D$

We conduct the training on ANN ResNet-50 for both methods, the results are shown in Tab. 5. It is

Table 5: Comparison of threshold schedulers when applied to ResNet-50 on ImageNet.

| Scheduler | STR ($D = 10^{-3}$) | | STDS ($D = 0.5$) | |
|---|---|---|---|---|
| | Sparsity (%) | Top-1 Acc. (%) | Sparsity (%) | Top-1 Acc. (%) |
| Sine | 94.14 | 71.51 | 95.72 | 72.42 |
| Linear | 95.41 | 69.85 | 98.46 | 59.79 |
| Log2 | 95.95 | 68.55 | 98.05 | 64.74 |

obvious that even though the final thresholds are set equally, the accuracy and overall sparsity vary with the scheduler. In fact, simply setting threshold trainable indicates one only cares about whether the final threshold is optimal, which turns out to be a tangential issue in soft threshold pruning.

The above results suggest the correct manner to manipulate threshold is pursuing a well-performed scheduler. Even if one studies the learning of threshold, it should be concentrated on the optimization of threshold scheduler, which may require tools in discrete-time optimal control. Due to the complex coupling between constantly changing threshold and final performance, discussion based on discrete-time optimal control is beyond the scope of this work.

## D DETAILED DERIVATION FOR LATS AND S-LATS

In this part, we provide detailed derivations for LATS and S-LATS, which covers Eq. 12 and Eq. 13.

## D.1 LATS FOR COSINE ANNEALING SCHEDULER

In most of the deep learning applications, the training process includes the schedule of the learning rate, which is also known as *learning rate scheduler*. Generally, the learning rate is updated at the end of an epoch. Assuming there are $N$ training epochs in total, each of which includes $B$ training mini-batches. The learning rate scheduler is defined as $\eta^{(n,b)} = h(n/N)$, which evaluates the learning rate at the $b$-th mini-batch in the $n$-th epoch. Here $n = 0, 1, \ldots, N-1$ denotes the current id of the training epoch, and $b = 1, 2, \ldots, B$ denotes the batch id in $n$-th epoch. We denote $h : [0, 1] \to \mathbb{R}^+$ as the scheduler function for the learning rate. For cosine annealing scheduler with $\eta_{\min} = 0$, we have

$$h(x) = \frac{\eta_{\max}}{2}(1 + \cos(\pi x)) \tag{28}$$

In Corollary 1, the threshold scheduler for LATS is obtained by $d^{(t)} = d^{(0)} + \mu \sum_{i=0}^{t-1} \eta^{(i)}$, where $d^{(t)}$ is the threshold after $t$ mini-batches from the beginning. Under the learning rate scheduler described in Eq. 28, the threshold $d^{(n,b)}$ is shown by accumulating all previous learning rates, which gives

$$d^{(n,b)} = d^{(0,0)} + \mu \left[ b \cdot h(n/N) + B \sum_{i=0}^{n-1} h(i/N) \right] \tag{29}$$

For cosine annealing learning rate scheduler, we have

$$
\begin{aligned}
d^{(n,b)} &= d^{(0,0)} + \frac{\mu\eta_{\max}}{2} \left[ b(1 + \cos\frac{n\pi}{N}) + B \sum_{i=0}^{n-1}(1 + \cos\frac{i\pi}{N}) \right] \\
&= d^{(0,0)} + \frac{\mu\eta_{\max}}{2} \left[ b(1 + \cos\frac{n\pi}{N}) + Bn + B \sum_{i=0}^{n-1} \cos\frac{i\pi}{N} \right]
\end{aligned}
\tag{30}
$$

To evaluate the sum of $\cos(i\pi/N)$, we give the following results

$$
\begin{aligned}
\sum_{i=0}^{n-1} \cos\frac{i\pi}{N} &= \frac{1}{\sin\frac{\pi}{2N}} \sum_{i=0}^{n-1} \left( \cos\frac{i\pi}{N} \sin\frac{\pi}{2N} \right) \\
&= \frac{1}{\sin\frac{\pi}{2N}} \sum_{i=0}^{n-1} \frac{1}{2} \left( \sin\left(\frac{i}{N} + \frac{1}{2N}\right)\pi - \sin\left(\frac{i}{N} - \frac{1}{2N}\right)\pi \right) \\
&= \frac{1}{2\sin\frac{\pi}{2N}} \left( \sum_{i=0}^{n-1} \sin\left(\frac{2i+1}{2N}\pi\right) - \sum_{i=0}^{n-1} \sin\left(\frac{2i-1}{2N}\pi\right) \right) \\
&= \frac{1}{2\sin\frac{\pi}{2N}} \left( \sum_{i=1}^{n} \sin\left(\frac{2i-1}{2N}\pi\right) - \sum_{i=0}^{n-1} \sin\left(\frac{2i-1}{2N}\pi\right) \right) \\
&= \frac{1}{2\sin\frac{\pi}{2N}} \left( \sin\left(\frac{2n-1}{2N}\pi\right) + \sin\frac{\pi}{2N} \right) \\
&= \frac{1}{2} + \frac{\sin\left(\frac{2n-1}{2N}\pi\right)}{2\sin\frac{\pi}{2N}}.
\end{aligned}
\tag{31}
$$

Recall Eq. 30, we have

$$
\begin{aligned}
d^{(n,b)} &= d^{(0,0)} + \frac{\mu\eta_{\max}}{2} \left[ b(1 + \cos\frac{n\pi}{N}) + Bn + B\left( \frac{1}{2} + \frac{\sin\left(\frac{2n-1}{2N}\pi\right)}{2\sin\frac{\pi}{2N}} \right) \right] \\
&= d^{(0,0)} + \mu\eta_{\max} \left[ \frac{b}{2}(1 + \cos\frac{n\pi}{N}) + \frac{B}{4}\left( 2n + 1 + \frac{\sin\left(\frac{2n-1}{2N}\pi\right)}{\sin\frac{\pi}{2N}} \right) \right].
\end{aligned}
\tag{32}
$$

When $d^{(0,0)} = 0$, this gives LATS in the form of Eq. 12.

## D.2 THE DETAILED MOTIVATION OF S-LATS

The implementation of LATS is rather complicated and requires meticulous coding. Worse still, for some learning rate schedulers, *e.g.*, polynomial decay scheduler (Liu et al., 2015; Chen et al., 2018)

$$\eta^{(t)} = \eta_{\max} \left( 1 - \frac{t}{T} \right)^{\kappa}, \tag{33}$$

where $\kappa > 0$ is a constant ($\kappa = 0.9$ in aforementioned studies), the form of LATS cannot be reduced like cosine annealing scheduler in most cases. To see this, we write the corresponding LATS as

$$d^{(t)} = d^{(0)} + \mu \eta_{\max} \sum_{i=0}^{t-1} \left( 1 - \frac{i}{T} \right)^{\kappa}. \tag{34}$$

Note that $1 - \frac{i}{T}$ makes up an arithmetic progression, and the threshold is the sum of their powers. Simplifying the sums of powers of arithmetic progression requires the so-called *Bernoulli number* (Jacobi, 1834; Knuth, 1993) when $\kappa$ is integer. For a general $\kappa > 0$, the expression of Eq. 34 includes the *generalized harmonic numbers* $H_n^{(-p)} := \sum_{k=1}^{n} k^p$, which is further based on the *Hurwitz zeta function* (Coffey, 2008). In such case, we cannot analytically compute LATS, which forces us to do the summation in Eq. 34. It thus brings about the accumulative error. In brief, we cannot expect each learning rate scheduler corresponds to an analytical and simple form of LATS.

To handle this, we turn to an approximation rather than precisely evaluating $d^{(t)}$. Returning to Eq. 29, with $d^{(0,0)}$ ignored, we split the right hand side into two terms $B \sum_{i=0}^{n-1} h(i/N)$ and $b \cdot h(n/N)$.

The first term could be viewed as left Riemann sum in two steps 1) interpolate points $0, 1/N, 2/N, \ldots, (n-1)/N, n/N$ with constant spacing $1/N$, the width of the rectangles 2) evaluate the sum of rectangle areas with height $h(i/N)$. It leads to the approximation of the Riemann integral

$$B \sum_{i=0}^{n-1} h(i/N) = BN \sum_{i=0}^{n-1} \frac{1}{N} h(i/N) \approx BN \int_{0}^{n/N} h(x) \mathrm{d}x. \tag{35}$$

The second term match $b/B$ part of a residual tiny rectangle

$$b \cdot h(n/N) = BN \cdot \frac{b}{B} \cdot \frac{1}{N} h(n/N) \tag{36}$$

To sum up, Eq. 35 and Eq. 36 together make a numerical approximation of integral $BN \int_{0}^{n/N+b/BN} h(x) \mathrm{d}x$, which is shown schematically in Fig. 5. Note that we could write training progress as $t/T$ now, where $t = Bn + b$ is the current iteration id and $T = BN$ is the total training iterations.

So far, we successfully replace the summation with integration. However, the current final threshold is $BN \int_{0}^{1} h(x) \mathrm{d}x$, which is different from the real one $d^{(N-1,B)}$. To keep the final threshold $d^{(T)} = d^{(N-1,B)}$, we normalize the integral as follows

$$d^{(t)} = d^{(N-1,B)} \cdot \frac{\int_{0}^{\frac{t}{T}} h(x) \mathrm{d}x}{\int_{0}^{1} h(x) \mathrm{d}x}. \tag{37}$$

For most learning rate scheduler functions, integration is much easier than summation. S-LATS enables us to apply our pruning method on wider varieties of deep learning applications.

## E  PRUNING EXPERIMENTS ON MOBILENET-V1 USING S-LATS

Besides the ResNet-like structures mentioned in the main text, we also conduct experiments on MobileNet-V1 (Howard et al., 2017) to show the power of our proposed methods S-LATS on the lightweight network. To make a fair comparison, we choose those SOTA studies using the standard

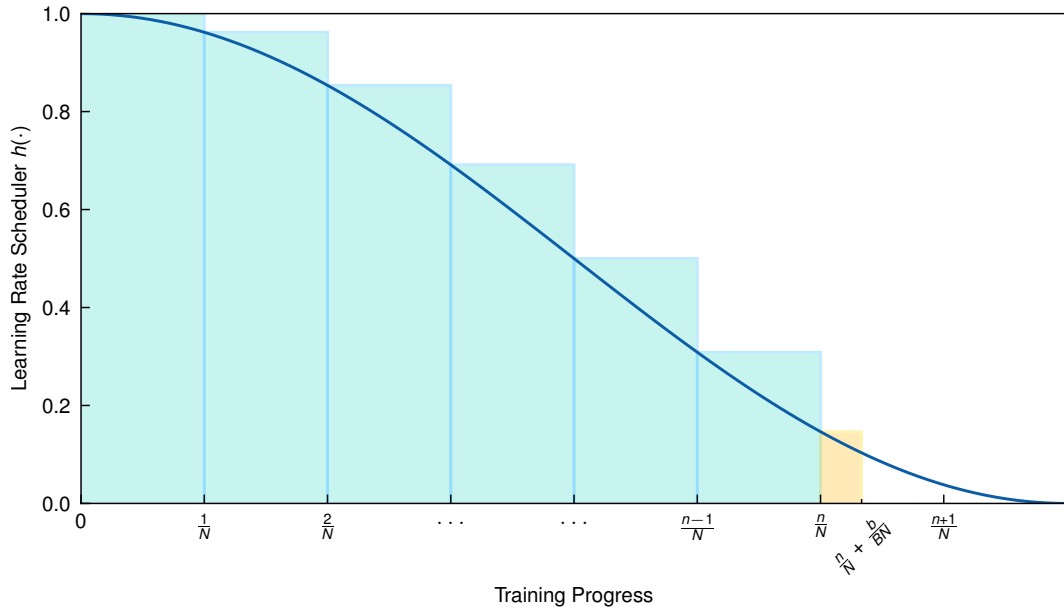

Figure 5: Explanation on numerical approximation of integral $\int_0^{n/N+b/BN} h(x)\mathrm{d}x$. Cosine annealing learning rate scheduler is exemplified in a darker blue curve. The area of $n$ cyan rectangles corresponds to Eq. 35. The area of the tiny yellow rectangle is Eq. 36.

training setting, *i.e.*, batch size of 256 and 100 training epochs. They include STR (Kusupati et al., 2020), gradual pruning in WoodFisher (Singh & Alistarh, 2020), and a modern implementation (Gale et al., 2019) of Gradual Magnitude Pruning (GMP) (Zhu & Gupta, 2017). Note that we do not compare S-LATS to OptG, since OptG adopts $1.8\times$ (180) training epochs, or the comparison would be unfair.

The results are shown in Tab. 6. Apparently, our proposed method towers over previous sparsify-during-training work. We elaborate on the sparsity budgets and the corresponding final thresholds in Tab. 12 of Appendix I. The hyperparameters for training MobileNet-V1 are stated in Tab. 8 of Appendix H.

Table 6: Performance comparison of MobileNet-V1 on ImageNet using standard training setting (256 batch size, 100 epochs). The results of GMP are gleaned from the manuscripts of STR and OptG. The accuracy of our method is averaged over three trials.

| Method | Top-1 Acc. (%) | Sparsity (%) |
|---|---|---|
| Dense | 71.95 | 0 |
| GMP | 67.70 | 74.11 |
| STR | 68.35 | 75.28 |
| STR | 66.52 | 79.07 |
| WoodFisher | 70.09 | 75.28 |
| **S-LATS** | **68.25**$_{\pm 0.19}$ | **81.84** |
| GMP | 61.80 | 89.03 |
| STR | 64.83 | 85.80 |
| STR | 62.10 | 89.01 |
| STR | 61.51 | 89.62 |
| WoodFisher | 63.87 | 89.00 |
| **S-LATS** | **66.73**$_{\pm 0.08}$ | **85.87** |
| **S-LATS** | **65.63**$_{\pm 0.20}$ | **88.22** |
| **S-LATS** | **64.93**$_{\pm 0.21}$ | **89.08** |

## F ABLATION STUDY OF THRESHOLD SCHEDULER ON RESNET-50

### F.1 REMOVE $L_2$ DECAY FOR FAIR COMPARISON

To evaluate the performance gains brought by the threshold scheduler alone, the *weight decay* or $L_2$ *penalty* must be removed. To explain this, recall line 4 in Algorithm 1, we know the update rule of hidden weight $\boldsymbol{\theta}$ is affected by both gradient and $L_2$ penalty $\lambda\|\boldsymbol{\theta}\|_2$. However, the analysis in Lemma 1 is based on vanilla SGD without weight decay. To account for this inconsistency, let's first investigate the influence of weight decay on the equivalent optimization problem.

In the presence of weight decay, the update rule described by Eq. 17 has an additional penalty term as follows

$$\boldsymbol{\theta}^{(t+1)} \leftarrow \boldsymbol{\theta}^{(t)} - \eta^{(t)}\nabla_{\boldsymbol{w}}(\mathcal{L}(\boldsymbol{w}^{(t)})) \odot \nabla_{\boldsymbol{\theta}}(\boldsymbol{w}(\boldsymbol{\theta}^{(t)}, d^{(t)})) - \eta^{(t)}\lambda\boldsymbol{\theta}^{(t)} \tag{38}$$

Following derivation in Eq. 20, we have

$$\theta^{(t+1)} = w^{(t)} + d^{(t)}\operatorname{sign}(w^{(t)}) - \eta^{(t)}\nabla_w(\mathcal{L}(w^{(t)})) - \eta^{(t)}\lambda\theta^{(t)} \tag{39}$$

Similarly, we denote $\bar{w}^{(t+1)} := w^{(t)} - \eta^{(t)}\nabla_w(\mathcal{L}(w^{(t)}))$ to be the target point of vanilla SGD without regularization. With Eq. 39, we have

$$
\begin{aligned}
\bar{w}^{(t+1)} &= \theta^{(t+1)} - d^{(t)}\operatorname{sign}(w^{(t)})\eta^{(t)} - \lambda\theta^{(t)} \\
&= \operatorname{sign}(\theta^{(t+1)})|\theta^{(t+1)}| - \operatorname{sign}(\theta^{(t)})d^{(t)} - \lambda\operatorname{sign}(\theta^{(t)})|\theta^{(t)}| \\
&= \operatorname{sign}(\theta^{(t+1)})|\theta^{(t+1)}| - \operatorname{sign}(\theta^{(t+1)})d^{(t)} - \lambda\operatorname{sign}(\theta^{(t)})|\theta^{(t)}| \\
&= \operatorname{sign}(\theta^{(t+1)})(|\theta^{(t+1)}| - d^{(t)} - \lambda|\theta^{(t)}|),
\end{aligned}
\tag{40}
$$

If $\bar{w}^{(t+1)}$ still satisfies the local relation that $\operatorname{sign}(w^{(t+1)}) = \operatorname{sign}(\theta^{(t+1)}) = \operatorname{sign}(\theta^{(t)}) = \operatorname{sign}(w^{(t)}) = \operatorname{sign}(\bar{w}^{(t+1)})$, mimicking Eq. 22, we have

$$w^{(t+1)} = \mathcal{S}_{d^{(t+1)} - d^{(t)} + \lambda|\theta^{(t)}|}(w^{(t)} - \eta^{(t)}\nabla_w(\mathcal{L}(w^{(t)}))). \tag{41}$$

Apparently, the $L_2$ penalty of $\boldsymbol{\theta}$ lies into the equivalent $L_1$ penalty term of $\boldsymbol{w}$ in the ISTA rule, making the analysis of the corresponding threshold scheduler intractable. Accordingly, we decide to remove weight decay, *i.e.*, and set $\lambda$ to zero in the ablation study to prevent an unpredictable threshold scheduler.

### F.2 SINE SCHEDULER IN STDS VS S-LATS

After removing weight decay, we rerun the original STDS and our methods under several sparsity levels (through changing $D$) while keeping the other hyperparameters and the batch size of 256. As illustrated in Fig. 6, the results on ResNet-50 show our method clearly surpasses the original STDS on the ImageNet dataset. A theoretical analysis is enclosed in Appendix B.

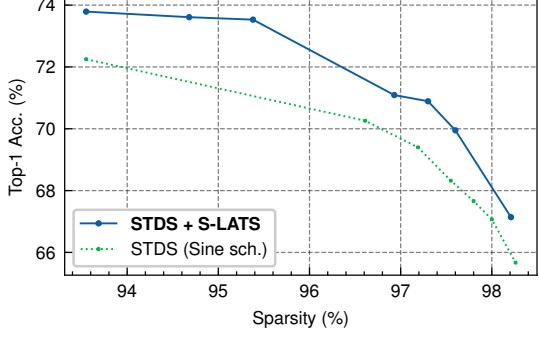

Figure 6: Performance comparison of original STDS and our method.

## G  SPARSITY VS FIRING RATE IN SNNS

### G.1  AN OVERVIEW OF SNNS

Spiking neural networks (SNNs) are honored as the third generation of neural network models (Maass, 1997), derived from biological neural network modeling. SNNs are composed of spiking neurons, which release spikes in binary form, and connections between neurons. The model of spiking neurons is a dynamical system described by one or more ordinary differential equations (ODE) and a firing threshold. The dynamical system is also called "subthreshold dynamics" in the context of computational neuroscience. A spike is generated and passed to all postsynaptic spiking neurons when the variable representing *membrane potential* exceeds the firing threshold. Today, the most commonly used neuron model is the *Leaky Integrate-and-Fire* (LIF) model. Specifically, LIF has the subthreshold dynamic as follows

$$\tau_m \frac{\mathrm{d}u(t)}{\mathrm{d}t} = -(u(t) - u_{\text{rest}}) + \sum Iw, \tag{42}$$

where $u(t)$ is the membrane potential at time $t$, $u_{\text{rest}}$ is the resting potential, $\tau_m$ is the membrane constant, $I$ and $w$ denote input spikes and input weights respectively. The firing behavior of LIF neurons is depicted as an instantaneous jump of membrane potential shown below

$$\lim_{\Delta t \to 0^+} u(t^f + \Delta t) = u_{\text{rest}}, \text{ if } u(t^f) \geq u_{\text{th}}, \tag{43}$$

where $u_{\text{th}}, t^f$ are the firing threshold and firing time respectively.

The ODE in Eq.42 can be discretized via the Euler method and transformed into an RNN-like iterative computing manner as follows

$$\begin{aligned}
u[t^-] &= u[t-1] + \frac{1}{\tau_m}\left(-(u[t-1] - u_{\text{rest}}) + \sum_i w_i I_i[t]\right), \\
s[t] &= H(u[t^-] - u_{\text{th}}), \\
u[t] &= s[t]u_{\text{rest}} + (1 - s[t])u[t^-].
\end{aligned} \tag{44}$$

where $u[t^-], u[t]$ are the membrane potential before and after firing at timestep $t$ respectively, $H(\cdot)$ is the Heaviside step function modeling jump behavior when a spike is triggered. However, training techniques in RNN, such as backpropagation through time (BPTT) (Werbos, 1990) cannot be directly applied to SNNs for the spiking behavior described by Heaviside is non-differentiable. Thanks to the *surrogate gradient* method proposed in Wu et al. (2018); Neftci et al. (2019), researchers can now incorporate BPTT into training of SNNs by switching to a "differentiable mode" of the Heaviside step function when computing gradient. It refers to replacing Heaviside step with a differentiable surrogate function. Surrogate gradient resembles the straight-through estimator (Bengio et al., 2013) closely in both computing style and ideology.

### G.2  REDUCING SNNS COST ON NEUROMORPHIC HARDWARE

SNNs are considered energy efficient when deployed on a series of dedicated hardware, also known as *neuromorphic hardware* or *event-driven hardware*. On these chips, the computation is triggered only when there are incoming spikes and weights are nonzero (Merolla et al., 2014). For this reason, there are three mainstream methods for alleviating the energy cost of a given SNN on neuromorphic chips including 1) unstructured pruning of weights, 2) reducing the number of spikes, and 3) searching for efficient SNN structures. The NAS studies on SNNs (Na et al., 2022; Kim et al., 2022a) are not based on existing SNN structure, so we omit the discussion of NAS methods.

Many recent studies have made ample signs of progress on pruning and reducing spike counts. They confirm there is a weak correlation between the number of spikes and weight sparsity (Deng et al., 2021; Chen et al., 2022; Kim et al., 2022b). We also evaluate the spike counts of pruned SNNs. Compared to the number of spikes, a more frequently used metric is *average firing rate*, which is obtained by averaging the number of spikes across timesteps and spiking neurons. We collect the average firing rate of each trial during inference using pruned SEW ResNet-18. We further provide a plot of the average firing rate against the sparsity, which is shown in Fig. 7. The weak relationship between the number of spikes and weight sparsity is manifested in the slightly decreased average

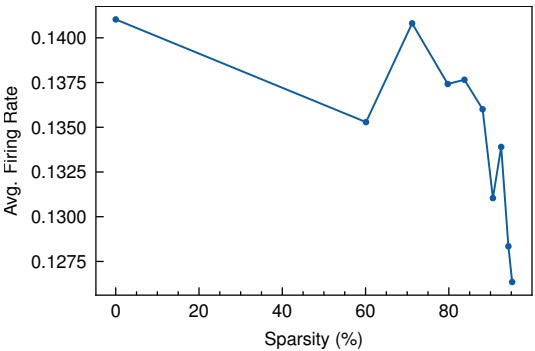

Figure 7: The trend of average firing rate against sparsity.

firing rate. Despite a downward trend in the average firing rate, the relative magnitude of the decline is trifling. It is consistent with previous observations and suggests pruning is an inefficient means of reducing the number of spikes in SNNs.

In conclusion, pruning is an efficient way to induce weight sparsity and lower cost. However, we should not expect the suppression of firing rates as a bonus.

## H  TRAINING HYPERPARAMETERS

We make the detailed setting in our experiments clear in Tab. 7, Tab. 8 and Tab. 9.

Table 7: ANN ResNet-50 hyperparameters.

| Description | Notation | Value |
| --- | --- | --- |
| # Epoch | - | 100 |
| Optimizer | - | Momentum SGD ($\mathrm{momentum} = 0.875$) |
| Overall batch size | - | 256    1024 |
| Max. learning rate | $\eta_{\max}$ | 0.256    0.512 |
| Learning rate scheduler | - | Cosine annealing |
| Warmup epochs | - | 5 |
| Label smoothing | - | 0.1 |
| Weight decay | $\lambda$ | 3.05e-5 (0 for ablation study in Appendix F) |
| Prune BN layers? | - | No |
| Prune first and last layers? | - | Yes |

Table 8: ANN MobileNet-V1 hyperparameters.

| Description | Notation | Value |
| --- | --- | --- |
| # Epoch | - | 100 |
| Optimizer | - | Momentum SGD ($\mathrm{momentum} = 0.875$) |
| Overall batch size | - | 256 |
| Max. learning rate | $\eta_{\max}$ | 0.256 |
| Learning rate scheduler | - | Cosine annealing |
| Warmup epochs | - | 5 |
| Label smoothing | - | 0.1 |
| Weight decay | $\lambda$ | 3.05e-5 |
| Prune BN layers? | - | No |
| Prune first and last layers? | - | Yes |

Table 9: SNN SEW ResNet-18 hyperparameters.

| Description | Notation | Value |
|---|---|---|
| # Epoch | - | 320 |
| Optimizer | - | Momentum SGD (momentum = 0.9) |
| Overall batch size | - | 256 |
| Max. learning rate | $\eta_{max}$ | 0.1 |
| Learning rate scheduler | - | Cosine annealing |
| Weight decay | $\lambda$ | 0 |
| Prune BN layers? | - | No |
| Prune first and last layers? | - | Yes |
| Simulation timesteps | - | 4 |
| SEW function | - | ADD |

# I SPARSITY BUDGETS

Table 10: Sparsity budgets of ResNet-50 using STDS + S-LATS on ImageNet (256 batch size).

| Final threshold D | 0.1 | 0.2 | 0.3 | 0.4 | 0.45 | 0.5 | 0.6 | 0.7 | 0.8 | 1.0 | 1.4 | 1.5 | 2.0 | 3.0 | 3.5 | 4.0 | 5.0 | 6.0 |
|---|---|---|---|---|---|---|---|---|---|---|---|---|---|---|---|---|---|---|
| Top-1 Acc. (%) | 76.75 | 75.52 | 74.50 | 73.75 | 73.18 | 73.03 | 72.04 | 71.47 | 70.81 | 69.64 | 67.47 | 67.02 | 64.20 | 60.35 | 58.88 | 56.79 | 53.64 | 50.41 |
| Layer(s) | | | | | | | | | Sparsity (%) | | | | | | | | | |
| Overall | 79.95 | 89.57 | 92.99 | 94.60 | 95.12 | 95.53 | 96.13 | 96.58 | 96.94 | 97.43 | 98.01 | 98.11 | 98.48 | 98.89 | 99.02 | 99.13 | 99.28 | 99.39 |
| conv1 | 34.18 | 49.83 | 56.73 | 59.57 | 64.35 | 67.90 | 69.79 | 70.89 | 75.54 | 74.31 | 78.36 | 78.71 | 81.92 | 83.94 | 85.58 | 86.20 | 88.01 | 89.47 |
| layer1.0.conv1 | 43.04 | 58.79 | 66.58 | 68.82 | 72.22 | 74.90 | 75.44 | 76.03 | 79.30 | 82.25 | 84.69 | 83.84 | 88.06 | 91.33 | 89.72 | 91.77 | 93.53 | 94.26 |
| layer1.0.conv2 | 73.59 | 82.82 | 87.36 | 89.43 | 90.82 | 91.38 | 92.01 | 93.18 | 93.06 | 94.96 | 95.97 | 95.87 | 97.08 | 98.22 | 97.96 | 98.44 | 98.51 | 99.00 |
| layer1.0.conv3 | 67.26 | 77.89 | 84.59 | 86.93 | 88.71 | 89.52 | 91.52 | 92.95 | 93.19 | 95.28 | 95.83 | 95.84 | 97.19 | 98.13 | 98.12 | 98.55 | 98.49 | 99.16 |
| layer1.0.downsample.0 | 58.61 | 73.39 | 78.28 | 82.06 | 83.91 | 85.33 | 85.86 | 87.01 | 89.11 | 90.31 | 91.83 | 92.61 | 93.19 | 94.56 | 95.14 | 95.73 | 96.19 | 96.62 |
| layer1.1.conv1 | 69.42 | 78.74 | 83.92 | 86.19 | 87.51 | 88.46 | 90.94 | 91.72 | 93.46 | 93.51 | 94.32 | 94.32 | 96.12 | 97.20 | 96.99 | 98.46 | 98.84 | 99.05 |
| layer1.1.conv2 | 73.06 | 84.09 | 88.31 | 89.18 | 90.32 | 91.81 | 93.83 | 94.73 | 96.02 | 95.62 | 95.72 | 95.36 | 97.18 | 97.96 | 98.02 | 99.20 | 99.64 | 99.41 |
| layer1.1.conv3 | 69.48 | 79.34 | 86.22 | 85.91 | 88.12 | 92.01 | 92.58 | 92.60 | 95.00 | 94.74 | 95.12 | 95.20 | 97.07 | 98.32 | 98.66 | 98.96 | 99.72 | 99.58 |
| layer1.2.conv1 | 64.66 | 75.52 | 82.27 | 86.07 | 87.00 | 87.75 | 89.77 | 88.64 | 91.03 | 92.55 | 94.18 | 95.09 | 96.27 | 97.03 | 97.97 | 97.82 | 98.78 | 98.91 |
| layer1.2.conv2 | 63.14 | 76.05 | 84.24 | 86.97 | 88.20 | 89.35 | 90.86 | 90.06 | 91.91 | 93.02 | 94.81 | 94.34 | 96.93 | 97.15 | 98.01 | 97.94 | 98.56 | 99.25 |
| layer1.2.conv3 | 68.90 | 80.36 | 86.72 | 87.87 | 89.11 | 90.09 | 91.77 | 92.69 | 92.73 | 94.42 | 96.64 | 95.80 | 97.64 | 98.26 | 99.04 | 98.71 | 99.18 | 99.33 |
| layer2.0.conv1 | 59.82 | 73.85 | 80.93 | 82.27 | 83.13 | 86.86 | 89.47 | 89.19 | 90.64 | 91.19 | 93.37 | 92.78 | 95.23 | 96.01 | 97.02 | 97.26 | 97.98 | 98.43 |
| layer2.0.conv2 | 72.90 | 85.04 | 88.93 | 92.21 | 92.57 | 93.19 | 94.54 | 94.80 | 94.89 | 95.74 | 97.05 | 97.01 | 97.50 | 98.04 | 98.27 | 98.28 | 98.61 | 99.16 |
| layer2.0.conv3 | 71.13 | 82.92 | 86.98 | 89.32 | 90.37 | 91.58 | 92.63 | 93.15 | 93.37 | 94.38 | 96.03 | 96.03 | 96.76 | 97.78 | 98.00 | 97.85 | 98.12 | 98.72 |
| layer2.0.downsample.0 | 81.42 | 90.05 | 92.85 | 94.08 | 94.78 | 95.22 | 95.66 | 96.28 | 96.44 | 97.40 | 97.88 | 98.07 | 98.25 | 98.86 | 99.03 | 99.08 | 99.16 | 99.31 |
| layer2.1.conv1 | 83.44 | 90.47 | 93.48 | 95.35 | 95.96 | 95.80 | 96.62 | 96.73 | 96.73 | 97.95 | 98.25 | 98.61 | 98.89 | 99.04 | 99.21 | 99.42 | 99.27 | 99.56 |
| layer2.1.conv2 | 82.60 | 90.41 | 94.05 | 96.28 | 96.22 | 96.37 | 97.17 | 96.90 | 97.76 | 98.31 | 98.68 | 99.02 | 99.14 | 99.44 | 99.46 | 99.61 | 99.56 | 99.76 |
| layer2.1.conv3 | 73.53 | 83.15 | 88.32 | 92.18 | 91.63 | 92.38 | 93.93 | 93.80 | 95.03 | 96.08 | 96.79 | 97.37 | 98.29 | 98.66 | 99.04 | 99.28 | 99.04 | 99.48 |
| layer2.2.conv1 | 74.48 | 85.75 | 88.91 | 91.48 | 92.37 | 91.97 | 93.44 | 94.26 | 94.86 | 95.85 | 96.09 | 96.88 | 97.54 | 98.33 | 98.46 | 98.97 | 98.85 | 99.08 |
| layer2.2.conv2 | 75.42 | 87.51 | 90.66 | 92.41 | 93.18 | 92.90 | 93.92 | 94.89 | 95.26 | 96.32 | 96.25 | 97.47 | 97.96 | 98.61 | 98.36 | 98.88 | 99.18 | 99.19 |
| layer2.2.conv3 | 70.68 | 82.69 | 86.23 | 90.11 | 90.35 | 91.14 | 92.36 | 93.93 | 93.28 | 95.32 | 96.33 | 96.91 | 97.50 | 98.33 | 98.39 | 98.82 | 99.04 | 99.12 |
| layer2.3.conv1 | 71.05 | 83.26 | 86.70 | 90.06 | 90.17 | 91.70 | 92.47 | 92.75 | 93.46 | 94.34 | 95.50 | 96.47 | 96.95 | 97.47 | 97.94 | 98.46 | 98.45 | 98.79 |
| layer2.3.conv2 | 74.50 | 83.83 | 88.23 | 91.11 | 91.28 | 92.90 | 93.06 | 94.27 | 94.36 | 94.89 | 97.02 | 96.67 | 97.29 | 97.94 | 98.25 | 98.73 | 98.92 | 98.61 |
| layer2.3.conv3 | 72.81 | 85.69 | 89.03 | 91.17 | 92.36 | 92.16 | 93.69 | 94.37 | 95.23 | 95.33 | 96.74 | 97.07 | 97.90 | 98.27 | 98.73 | 99.03 | 99.11 | 99.04 |
| layer3.0.conv1 | 61.26 | 74.70 | 80.18 | 83.80 | 84.81 | 85.51 | 87.13 | 88.01 | 88.81 | 90.43 | 91.82 | 92.57 | 93.79 | 95.04 | 95.56 | 96.42 | 96.69 | 97.29 |
| layer3.0.conv2 | 82.41 | 91.42 | 94.51 | 95.78 | 96.15 | 96.49 | 96.94 | 97.27 | 97.56 | 97.94 | 98.33 | 98.33 | 98.68 | 98.87 | 99.04 | 99.12 | 99.25 | 99.35 |
| layer3.0.conv3 | 71.21 | 82.18 | 86.32 | 88.76 | 89.38 | 89.93 | 91.42 | 92.04 | 92.46 | 93.73 | 94.97 | 95.40 | 96.23 | 97.28 | 97.50 | 97.95 | 98.20 | 98.61 |
| layer3.0.downsample.0 | 86.21 | 93.20 | 95.39 | 96.83 | 96.96 | 97.29 | 97.77 | 97.98 | 98.13 | 98.53 | 98.83 | 98.96 | 99.24 | 99.39 | 99.44 | 99.56 | 99.60 | 99.60 |
| layer3.1.conv1 | 86.76 | 92.91 | 95.17 | 96.46 | 96.48 | 97.02 | 97.25 | 97.80 | 97.95 | 98.37 | 98.67 | 98.82 | 99.16 | 99.38 | 99.41 | 99.55 | 99.56 | 99.63 |
| layer3.1.conv2 | 86.67 | 93.68 | 95.71 | 96.79 | 97.09 | 97.37 | 97.77 | 98.09 | 98.32 | 98.52 | 98.92 | 98.99 | 99.31 | 99.51 | 99.51 | 99.66 | 99.67 | 99.72 |
| layer3.1.conv3 | 75.18 | 86.50 | 90.08 | 92.77 | 93.44 | 94.10 | 94.53 | 95.48 | 96.12 | 96.75 | 97.53 | 97.82 | 98.46 | 99.03 | 98.98 | 99.29 | 99.38 | 99.50 |
| layer3.2.conv1 | 83.83 | 90.82 | 93.96 | 95.29 | 96.06 | 95.69 | 97.01 | 96.80 | 97.13 | 97.58 | 98.36 | 98.28 | 98.65 | 99.08 | 99.32 | 99.27 | 99.55 | 99.58 |
| layer3.2.conv2 | 84.24 | 92.07 | 94.68 | 95.84 | 96.17 | 96.22 | 97.35 | 97.26 | 97.53 | 97.93 | 98.62 | 98.69 | 98.92 | 99.22 | 99.42 | 99.40 | 99.62 | 99.66 |
| layer3.2.conv3 | 75.66 | 85.49 | 89.88 | 92.30 | 93.13 | 92.99 | 95.21 | 94.99 | 95.49 | 96.24 | 97.61 | 97.65 | 98.28 | 98.83 | 99.14 | 99.10 | 99.44 | 99.54 |
| layer3.3.conv1 | 80.26 | 90.40 | 92.47 | 94.60 | 94.41 | 94.94 | 95.91 | 95.87 | 96.48 | 97.20 | 98.17 | 98.13 | 98.67 | 98.84 | 99.03 | 99.09 | 99.32 | 99.53 |
| layer3.3.conv2 | 83.78 | 91.46 | 94.33 | 95.39 | 96.14 | 96.59 | 96.90 | 97.70 | 97.88 | 98.34 | 98.75 | 98.97 | 99.12 | 99.40 | 99.49 | 99.55 | 99.62 | 99.70 |
| layer3.3.conv3 | 76.82 | 87.45 | 91.13 | 93.10 | 93.32 | 94.60 | 95.05 | 96.24 | 96.35 | 97.10 | 98.08 | 98.33 | 98.66 | 99.06 | 99.26 | 99.36 | 99.43 | 99.53 |
| layer3.4.conv1 | 77.62 | 87.58 | 91.12 | 92.47 | 93.41 | 94.15 | 94.67 | 96.03 | 96.16 | 96.44 | 97.39 | 97.55 | 98.16 | 98.85 | 98.84 | 99.19 | 99.38 | 99.37 |
| layer3.4.conv2 | 83.13 | 91.87 | 94.67 | 95.63 | 96.17 | 96.66 | 97.32 | 97.48 | 97.87 | 98.14 | 98.61 | 98.71 | 99.11 | 99.42 | 99.53 | 99.63 | 99.73 | 99.72 |
| layer3.4.conv3 | 76.09 | 87.14 | 91.38 | 92.55 | 93.79 | 94.55 | 95.29 | 95.94 | 96.44 | 97.00 | 97.93 | 98.10 | 98.52 | 99.14 | 99.23 | 99.43 | 99.62 | 99.62 |
| layer3.5.conv1 | 73.48 | 85.83 | 89.67 | 91.89 | 92.45 | 93.36 | 94.45 | 94.68 | 94.92 | 95.73 | 96.69 | 97.02 | 97.61 | 98.39 | 98.88 | 98.86 | 99.13 | 99.23 |
| layer3.5.conv2 | 81.39 | 90.78 | 93.61 | 95.30 | 95.85 | 96.28 | 96.62 | 97.39 | 97.56 | 98.09 | 98.60 | 98.59 | 99.00 | 99.33 | 99.53 | 99.50 | 99.61 | 99.69 |
| layer3.5.conv3 | 73.98 | 85.35 | 89.23 | 91.80 | 92.53 | 93.13 | 94.12 | 95.16 | 95.16 | 95.98 | 96.99 | 97.31 | 98.06 | 98.69 | 99.14 | 99.21 | 99.23 | 99.44 |
| layer4.0.conv1 | 64.48 | 77.49 | 82.89 | 85.92 | 87.00 | 87.93 | 89.26 | 90.33 | 91.14 | 92.09 | 93.65 | 93.85 | 94.79 | 95.83 | 96.25 | 96.68 | 97.19 | 97.40 |
| layer4.0.conv2 | 84.02 | 94.14 | 97.01 | 97.83 | 98.07 | 98.17 | 98.34 | 98.45 | 98.66 | 98.82 | 99.05 | 99.05 | 99.17 | 99.36 | 99.38 | 99.44 | 99.53 | 99.57 |
| layer4.0.conv3 | 72.53 | 83.17 | 87.02 | 89.51 | 90.39 | 90.99 | 92.21 | 93.09 | 93.73 | 94.63 | 95.87 | 96.01 | 96.73 | 97.61 | 97.78 | 98.02 | 98.38 | 98.60 |
| layer4.0.downsample.0 | 85.36 | 92.49 | 94.97 | 96.03 | 96.39 | 96.77 | 97.13 | 97.42 | 97.63 | 97.92 | 98.33 | 98.41 | 98.70 | 99.00 | 99.13 | 99.20 | 99.33 | 99.44 |
| layer4.1.conv1 | 81.34 | 90.10 | 93.30 | 94.91 | 95.55 | 95.99 | 96.52 | 97.17 | 97.47 | 98.00 | 98.60 | 98.59 | 98.86 | 99.23 | 99.35 | 99.42 | 99.56 | 99.65 |
| layer4.1.conv2 | 83.00 | 91.75 | 94.76 | 96.19 | 96.68 | 97.03 | 97.39 | 97.85 | 98.14 | 98.52 | 98.95 | 98.96 | 99.18 | 99.46 | 99.55 | 99.57 | 99.69 | 99.73 |
| layer4.1.conv3 | 75.37 | 86.50 | 90.65 | 92.84 | 93.42 | 94.08 | 94.81 | 95.43 | 95.88 | 96.63 | 97.40 | 97.48 | 97.79 | 98.47 | 98.59 | 98.64 | 98.88 | 99.00 |
| layer4.2.conv1 | 74.54 | 86.22 | 90.56 | 92.98 | 93.83 | 94.53 | 95.26 | 96.01 | 96.63 | 97.23 | 97.82 | 97.94 | 98.35 | 98.84 | 98.93 | 99.03 | 99.21 | 99.39 |
| layer4.2.conv2 | 79.66 | 90.89 | 95.74 | 97.04 | 97.38 | 97.52 | 97.81 | 98.07 | 98.36 | 98.56 | 98.77 | 98.82 | 99.01 | 99.23 | 99.29 | 99.38 | 99.49 | 99.60 |
| layer4.2.conv3 | 66.98 | 79.13 | 84.53 | 87.56 | 88.74 | 89.37 | 90.60 | 91.60 | 92.61 | 93.82 | 95.04 | 95.41 | 96.41 | 97.57 | 97.79 | 98.08 | 98.51 | 98.86 |
| fc | 86.44 | 94.47 | 96.69 | 97.58 | 97.87 | 98.10 | 98.44 | 98.65 | 98.79 | 99.04 | 99.25 | 99.27 | 99.40 | 99.49 | 99.54 | 99.56 | 99.57 | 99.60 |

Table 11: Sparsity budgets of ResNet-50 using STDS + S-LATS on ImageNet (1024 batch size).

| Final threshold $D$ | 0.1 | 0.2 | 0.23 | 0.25 | 0.3 | 0.4 | 0.475 | 0.5 | 0.6 | 0.73 | 0.8 | 1.0 | 1.13 | 1.5 | 2.0 | 3.0 | 3.5 | 4.0 |
|---|---|---|---|---|---|---|---|---|---|---|---|---|---|---|---|---|---|---|
| Top-1 Acc. (%) | 76.61 | 76.15 | 75.97 | 75.88 | 75.58 | 74.90 | 74.61 | 74.04 | 73.68 | 72.84 | 72.73 | 71.67 | 70.68 | 69.20 | 67.16 | 63.81 | 61.72 | 60.40 |
| Layer(s) | Sparsity (%) | | | | | | | | | | | | | | | | | |
| Overall | 79.00 | 88.81 | 90.15 | 90.92 | 92.34 | 94.19 | 95.01 | 95.25 | 95.93 | 96.53 | 96.78 | 97.30 | 97.54 | 98.00 | 98.38 | 98.79 | 98.93 | 99.04 |
| conv1 | 40.67 | 46.90 | 55.02 | 53.64 | 56.36 | 62.73 | 66.07 | 64.46 | 71.75 | 70.97 | 72.86 | 74.04 | 76.87 | 77.51 | 79.39 | 83.93 | 84.56 | 87.38 |
| layer1.0.conv1 | 52.32 | 58.42 | 65.16 | 64.92 | 65.36 | 72.02 | 74.71 | 74.07 | 79.30 | 79.30 | 81.84 | 82.93 | 85.82 | 83.20 | 87.96 | 90.33 | 91.04 | 90.97 |
| layer1.0.conv2 | 75.36 | 82.51 | 85.46 | 86.66 | 87.70 | 89.99 | 91.46 | 91.87 | 92.69 | 93.60 | 94.34 | 94.89 | 95.41 | 95.79 | 96.87 | 97.73 | 98.06 | 98.10 |
| layer1.0.conv3 | 70.77 | 77.27 | 80.81 | 81.59 | 83.83 | 86.68 | 88.84 | 89.89 | 90.94 | 92.82 | 93.27 | 94.27 | 94.79 | 95.66 | 96.89 | 97.58 | 97.92 | 98.32 |
| layer1.0.downsample.0 | 64.12 | 71.25 | 74.69 | 75.47 | 78.57 | 82.09 | 83.73 | 83.44 | 86.99 | 87.38 | 88.92 | 89.49 | 90.48 | 92.49 | 93.05 | 94.68 | 94.98 | 95.31 |
| layer1.1.conv1 | 74.98 | 79.80 | 82.61 | 82.99 | 85.29 | 86.96 | 87.48 | 89.42 | 90.72 | 91.63 | 92.41 | 94.35 | 93.60 | 95.78 | 96.36 | 97.36 | 97.49 | 98.77 |
| layer1.1.conv2 | 78.53 | 83.95 | 86.02 | 85.23 | 89.04 | 88.26 | 90.46 | 92.69 | 92.72 | 93.74 | 94.16 | 95.35 | 95.43 | 96.98 | 97.64 | 98.41 | 98.84 | 99.33 |
| layer1.1.conv3 | 77.44 | 80.13 | 81.32 | 81.99 | 84.70 | 85.31 | 88.08 | 90.27 | 91.71 | 93.14 | 93.58 | 94.20 | 94.73 | 96.28 | 96.94 | 98.08 | 98.66 | 99.07 |
| layer1.2.conv1 | 70.75 | 76.64 | 77.59 | 79.77 | 80.70 | 85.06 | 86.50 | 87.58 | 89.15 | 90.91 | 92.42 | 92.96 | 93.51 | 94.43 | 96.46 | 98.02 | 97.99 | 97.89 |
| layer1.2.conv2 | 70.35 | 76.21 | 78.69 | 80.13 | 83.24 | 86.20 | 86.69 | 88.74 | 91.18 | 91.75 | 92.65 | 94.47 | 93.40 | 95.38 | 97.15 | 98.01 | 97.97 | 98.34 |
| layer1.2.conv3 | 77.30 | 77.87 | 82.51 | 82.66 | 84.60 | 86.46 | 89.57 | 88.23 | 92.04 | 92.70 | 92.96 | 95.35 | 93.87 | 96.25 | 97.64 | 98.19 | 98.46 | 98.26 |
| layer2.0.conv1 | 69.29 | 72.99 | 77.38 | 75.72 | 79.36 | 81.06 | 85.20 | 85.80 | 88.93 | 88.83 | 90.87 | 92.46 | 92.42 | 94.80 | 95.63 | 96.78 | 97.29 | 97.39 |
| layer2.0.conv2 | 76.86 | 84.95 | 87.06 | 88.11 | 89.32 | 91.52 | 93.28 | 93.42 | 94.01 | 94.81 | 95.40 | 95.92 | 96.71 | 96.96 | 97.50 | 98.52 | 98.50 | 98.62 |
| layer2.0.conv3 | 75.99 | 80.48 | 84.57 | 85.12 | 85.54 | 88.53 | 90.73 | 90.84 | 91.45 | 92.50 | 93.01 | 94.82 | 94.87 | 95.76 | 97.05 | 97.80 | 97.77 | 98.13 |
| layer2.0.downsample.0 | 84.75 | 89.95 | 91.27 | 91.94 | 92.58 | 94.84 | 95.10 | 95.29 | 95.79 | 96.67 | 96.86 | 97.33 | 97.49 | 97.89 | 98.41 | 98.67 | 98.93 | 98.96 |
| layer2.1.conv1 | 89.09 | 90.01 | 91.85 | 93.14 | 92.72 | 94.67 | 95.55 | 95.54 | 95.65 | 97.00 | 97.05 | 97.73 | 98.01 | 98.23 | 98.93 | 99.25 | 99.44 | 99.39 |
| layer2.1.conv2 | 87.41 | 89.84 | 92.13 | 93.68 | 92.94 | 95.04 | 95.99 | 96.01 | 96.07 | 97.35 | 97.49 | 98.36 | 98.26 | 98.67 | 99.17 | 99.47 | 99.65 | 99.65 |
| layer2.1.conv3 | 81.25 | 82.44 | 86.02 | 88.49 | 87.05 | 90.50 | 92.51 | 92.16 | 91.95 | 94.20 | 94.48 | 95.95 | 96.12 | 97.09 | 97.89 | 98.80 | 99.16 | 99.10 |
| layer2.2.conv1 | 77.10 | 85.06 | 86.77 | 88.37 | 89.76 | 91.35 | 91.45 | 92.98 | 93.09 | 93.55 | 94.63 | 95.11 | 96.12 | 96.55 | 97.05 | 97.93 | 98.79 | 98.58 |
| layer2.2.conv2 | 79.02 | 87.80 | 88.76 | 89.76 | 89.97 | 92.55 | 92.77 | 92.61 | 93.93 | 94.61 | 95.09 | 95.26 | 96.40 | 97.14 | 97.50 | 98.37 | 99.17 | 98.94 |
| layer2.2.conv3 | 71.18 | 82.41 | 83.43 | 85.35 | 87.02 | 89.28 | 90.05 | 90.93 | 91.46 | 92.80 | 93.58 | 93.83 | 95.80 | 95.97 | 96.96 | 97.96 | 98.82 | 98.57 |
| layer2.3.conv1 | 71.29 | 81.69 | 84.35 | 85.66 | 87.46 | 89.64 | 90.34 | 90.71 | 91.64 | 93.72 | 93.07 | 94.49 | 95.54 | 95.78 | 97.13 | 97.82 | 97.49 | 98.08 |
| layer2.3.conv2 | 73.69 | 83.46 | 85.49 | 86.71 | 88.83 | 90.38 | 91.28 | 92.67 | 93.19 | 94.32 | 93.90 | 95.26 | 96.04 | 96.57 | 97.52 | 97.67 | 97.97 | 98.33 |
| layer2.3.conv3 | 73.48 | 84.16 | 86.39 | 86.28 | 89.51 | 90.65 | 92.20 | 92.08 | 93.02 | 94.78 | 94.69 | 95.57 | 95.97 | 96.05 | 97.56 | 98.05 | 98.25 | 98.75 |
| layer3.0.conv1 | 62.93 | 75.02 | 77.81 | 79.06 | 81.57 | 83.61 | 85.41 | 85.64 | 87.35 | 88.80 | 89.57 | 90.65 | 91.63 | 92.85 | 94.27 | 95.65 | 96.26 | 96.18 |
| layer3.0.conv2 | 82.70 | 91.71 | 92.62 | 93.54 | 94.69 | 95.64 | 96.15 | 96.70 | 96.96 | 97.45 | 97.57 | 97.90 | 98.01 | 98.43 | 98.70 | 98.96 | 99.07 | 99.11 |
| layer3.0.conv3 | 72.49 | 82.71 | 84.57 | 85.53 | 87.34 | 88.74 | 89.68 | 90.66 | 91.39 | 92.98 | 92.80 | 93.95 | 94.79 | 95.56 | 96.47 | 97.36 | 97.68 | 97.79 |
| layer3.0.downsample.0 | 87.88 | 93.93 | 94.74 | 95.39 | 96.36 | 96.89 | 97.35 | 97.51 | 97.93 | 98.35 | 98.38 | 98.69 | 98.76 | 99.06 | 99.25 | 99.36 | 99.44 | 99.49 |
| layer3.1.conv1 | 86.70 | 92.88 | 93.65 | 94.35 | 95.64 | 96.29 | 96.63 | 97.02 | 97.19 | 97.72 | 97.75 | 97.94 | 98.49 | 98.69 | 99.02 | 99.25 | 99.39 | 99.41 |
| layer3.1.conv2 | 85.98 | 93.44 | 93.99 | 94.73 | 96.13 | 96.67 | 97.06 | 97.37 | 97.48 | 98.13 | 98.19 | 98.35 | 98.87 | 98.90 | 99.11 | 99.34 | 99.48 | 99.43 |
| layer3.1.conv3 | 74.52 | 85.38 | 87.49 | 88.09 | 90.54 | 92.11 | 93.02 | 93.65 | 94.33 | 95.16 | 95.69 | 96.05 | 97.08 | 97.38 | 98.07 | 98.38 | 98.85 | 98.90 |
| layer3.2.conv1 | 84.36 | 91.82 | 93.20 | 93.33 | 93.93 | 95.07 | 95.61 | 96.23 | 96.26 | 97.09 | 97.40 | 97.48 | 98.08 | 98.13 | 98.50 | 99.02 | 99.16 | 99.13 |
| layer3.2.conv2 | 83.49 | 91.97 | 93.01 | 93.50 | 94.42 | 95.51 | 96.15 | 96.73 | 96.65 | 97.35 | 97.59 | 97.67 | 98.18 | 98.21 | 98.75 | 99.15 | 99.20 | 99.22 |
| layer3.2.conv3 | 75.10 | 85.78 | 87.87 | 88.06 | 89.36 | 91.66 | 92.72 | 93.96 | 94.17 | 95.06 | 95.61 | 95.91 | 96.77 | 96.89 | 98.07 | 98.41 | 98.71 | 98.83 |
| layer3.3.conv1 | 80.11 | 89.01 | 90.31 | 91.45 | 92.54 | 93.60 | 94.34 | 95.01 | 95.78 | 96.18 | 96.36 | 97.12 | 97.36 | 97.63 | 98.26 | 98.61 | 99.07 | 99.02 |
| layer3.3.conv2 | 82.74 | 91.07 | 92.45 | 92.82 | 93.74 | 95.54 | 96.15 | 96.43 | 96.95 | 97.39 | 97.85 | 98.00 | 98.54 | 98.79 | 99.22 | 99.37 | 99.33 | |
| layer3.3.conv3 | 75.19 | 86.40 | 87.38 | 88.85 | 89.98 | 92.03 | 93.14 | 93.50 | 94.62 | 95.83 | 95.34 | 96.32 | 96.50 | 97.57 | 98.04 | 98.73 | 98.93 | 98.85 |
| layer3.4.conv1 | 76.54 | 86.54 | 88.09 | 89.35 | 90.46 | 92.61 | 93.25 | 94.02 | 94.56 | 95.68 | 95.20 | 95.73 | 96.46 | 97.12 | 97.73 | 98.47 | 98.51 | 98.69 |
| layer3.4.conv2 | 82.23 | 90.86 | 92.02 | 92.50 | 93.93 | 95.47 | 96.05 | 95.86 | 96.63 | 97.20 | 97.72 | 97.97 | 97.90 | 98.59 | 98.88 | 99.21 | 99.24 | 99.40 |
| layer3.4.conv3 | 75.51 | 85.30 | 87.73 | 88.50 | 90.23 | 91.95 | 92.59 | 93.37 | 94.24 | 95.53 | 95.63 | 96.11 | 96.72 | 97.46 | 98.11 | 98.56 | 98.42 | 98.90 |
| layer3.5.conv1 | 73.52 | 84.35 | 86.23 | 86.85 | 89.11 | 91.01 | 91.75 | 92.27 | 93.27 | 94.56 | 94.89 | 95.45 | 95.75 | 96.62 | 97.41 | 98.11 | 98.50 | 98.62 |
| layer3.5.conv2 | 80.96 | 90.30 | 91.09 | 91.95 | 93.39 | 95.08 | 95.45 | 95.67 | 96.55 | 97.03 | 97.39 | 97.74 | 97.88 | 98.39 | 98.76 | 99.30 | 99.30 | 99.43 |
| layer3.5.conv3 | 73.55 | 83.72 | 85.38 | 86.40 | 88.65 | 90.83 | 91.40 | 91.96 | 93.12 | 94.61 | 94.68 | 95.12 | 95.74 | 96.61 | 97.57 | 98.10 | 98.47 | 98.61 |
| layer4.0.conv1 | 65.17 | 78.27 | 79.76 | 81.06 | 83.34 | 86.00 | 87.24 | 87.75 | 89.04 | 90.32 | 90.94 | 91.87 | 92.49 | 93.47 | 94.63 | 95.65 | 96.29 | 96.39 |
| layer4.0.conv2 | 84.07 | 93.29 | 94.58 | 95.35 | 96.59 | 97.99 | 98.29 | 98.30 | 98.51 | 98.68 | 98.75 | 98.92 | 98.97 | 99.09 | 99.21 | 99.37 | 99.41 | 99.45 |
| layer4.0.conv3 | 72.08 | 83.90 | 85.07 | 86.04 | 87.64 | 89.90 | 90.95 | 91.05 | 92.18 | 93.23 | 93.78 | 94.72 | 95.02 | 95.93 | 96.67 | 97.51 | 97.81 | 98.10 |
| layer4.0.downsample.0 | 85.95 | 92.56 | 93.61 | 94.12 | 96.16 | 96.57 | 96.80 | 97.16 | 97.64 | 97.66 | 98.01 | 98.22 | 98.54 | 98.83 | 99.08 | 99.23 | 99.24 | |
| layer4.1.conv1 | 77.97 | 89.09 | 89.96 | 91.06 | 92.44 | 94.43 | 94.94 | 95.23 | 95.96 | 96.52 | 96.91 | 97.39 | 97.68 | 98.12 | 98.49 | 98.88 | 99.01 | 99.18 |
| layer4.1.conv2 | 80.73 | 90.68 | 91.40 | 92.49 | 94.03 | 95.76 | 96.24 | 96.56 | 97.11 | 97.62 | 97.88 | 98.30 | 98.47 | 98.83 | 99.02 | 99.32 | 99.38 | 99.48 |
| layer4.1.conv3 | 71.47 | 85.15 | 86.25 | 87.73 | 89.75 | 92.46 | 92.96 | 93.54 | 94.38 | 95.09 | 95.65 | 96.33 | 96.71 | 97.38 | 97.58 | 98.12 | 98.23 | 98.56 |
| layer4.2.conv1 | 69.25 | 83.86 | 86.10 | 86.64 | 88.26 | 91.54 | 93.58 | 93.38 | 94.55 | 95.19 | 95.67 | 96.81 | 96.90 | 97.62 | 97.87 | 98.41 | 98.48 | 98.75 |
| layer4.2.conv2 | 75.76 | 87.45 | 89.41 | 89.79 | 91.75 | 94.72 | 94.72 | 96.34 | 97.50 | 97.70 | 98.03 | 98.46 | 98.47 | 98.83 | 98.96 | 99.27 | 99.29 | 99.39 |
| layer4.2.conv3 | 62.85 | 77.36 | 79.58 | 80.52 | 82.77 | 86.37 | 88.80 | 88.74 | 90.50 | 91.41 | 91.98 | 93.85 | 93.97 | 95.32 | 96.19 | 97.41 | 97.61 | 98.02 |
| fc | 88.02 | 95.57 | 96.39 | 96.77 | 97.50 | 98.29 | 98.54 | 98.65 | 98.92 | 99.16 | 99.20 | 99.35 | 99.41 | 99.52 | 99.62 | 99.69 | 99.69 | 99.65 |

Table 12: Sparsity budgets of MobileNet-V1 using STDS + S-LATS on ImageNet.

| Final threshold $D$ | 0.4 | 0.6 | 0.8 | 0.9 |
|---|---|---|---|---|
| Top-1 Acc. (%) | 68.44 | 66.64 | 65.41 | 65.13 |
| Layer(s) | Sparsity (%) | | | |
| Overall | 81.84 | 85.87 | 88.22 | 89.08 |
| model.0.0 | 51.39 | 62.27 | 57.06 | 55.90 |
| model.1.0 | 46.53 | 62.85 | 54.86 | 56.94 |
| model.1.3 | 60.45 | 76.76 | 67.63 | 68.51 |
| model.2.0 | 11.28 | 21.70 | 18.75 | 18.23 |
| model.2.3 | 52.82 | 61.65 | 61.56 | 68.92 |
| model.3.0 | 23.96 | 25.43 | 26.04 | 31.86 |
| model.3.3 | 53.38 | 59.64 | 64.90 | 67.77 |
| model.4.0 | 2.69 | 5.64 | 8.94 | 5.38 |
| model.4.3 | 62.50 | 68.58 | 72.97 | 75.10 |
| model.5.0 | 20.70 | 26.26 | 29.34 | 31.38 |
| model.5.3 | 68.64 | 74.57 | 78.17 | 79.69 |
| model.6.0 | 15.32 | 20.49 | 20.40 | 25.00 |
| model.6.3 | 79.08 | 84.26 | 87.14 | 87.85 |
| model.7.0 | 24.80 | 29.28 | 34.44 | 37.83 |
| model.7.3 | 81.65 | 86.53 | 89.26 | 90.23 |
| model.8.0 | 35.72 | 40.89 | 48.55 | 48.59 |
| model.8.3 | 81.10 | 85.35 | 88.22 | 89.20 |
| model.9.0 | 40.06 | 42.73 | 46.59 | 50.85 |
| model.9.3 | 79.38 | 84.02 | 87.30 | 87.97 |
| model.10.0 | 33.88 | 40.41 | 42.99 | 45.33 |
| model.10.3 | 75.01 | 80.66 | 83.84 | 85.23 |
| model.11.0 | 26.52 | 28.56 | 31.77 | 31.42 |
| model.11.3 | 71.13 | 77.06 | 80.75 | 82.36 |
| model.12.0 | 10.44 | 13.24 | 14.67 | 16.06 |
| model.12.3 | 80.64 | 84.83 | 87.28 | 88.15 |
| model.13.0 | 43.65 | 46.79 | 48.35 | 49.09 |
| model.13.3 | 82.10 | 85.47 | 87.57 | 88.36 |
| fc | 92.40 | 95.25 | 96.54 | 96.93 |

Table 13: Sparsity budgets of ResNet-50 using PGH scheduler in pruning at initialization setting on ImageNet.

| Final threshold $D$ | 0.1 | 0.11 | 0.13 | 0.15 |
|---|---|---|---|---|
| Top-1 Acc. (%) | 74.69 | 72.89 | 68.23 | 62.22 |
| Layer(s) | Sparsity (%) | | | |
| Overall | 87.16 | 90.00 | 93.11 | 95.64 |
| conv1 | 32.05 | 33.48 | 34.66 | 34.75 |
| layer1.0.conv1 | 38.92 | 38.70 | 37.48 | 33.89 |
| layer1.0.conv2 | 67.15 | 67.55 | 66.07 | 59.91 |
| layer1.0.conv3 | 59.89 | 60.16 | 57.60 | 49.34 |
| layer1.0.downsample.0 | 58.89 | 61.24 | 64.05 | 60.72 |
| layer1.1.conv1 | 60.78 | 63.04 | 61.00 | 53.19 |
| layer1.1.conv2 | 65.13 | 64.05 | 60.48 | 54.98 |
| layer1.1.conv3 | 57.58 | 55.88 | 49.53 | 42.51 |
| layer1.2.conv1 | 53.80 | 55.80 | 51.01 | 43.32 |
| layer1.2.conv2 | 58.37 | 59.34 | 55.33 | 51.22 |
| layer1.2.conv3 | 61.24 | 62.34 | 60.16 | 48.72 |
| layer2.0.conv1 | 49.99 | 49.88 | 45.61 | 42.38 |
| layer2.0.conv2 | 70.38 | 69.61 | 72.09 | 86.96 |
| layer2.0.conv3 | 63.37 | 61.13 | 59.27 | 82.79 |
| layer2.0.downsample.0 | 75.97 | 74.72 | 70.28 | 59.11 |
| layer2.1.conv1 | 77.21 | 75.13 | 66.76 | 59.39 |
| layer2.1.conv2 | 76.35 | 74.61 | 73.73 | 86.70 |
| layer2.1.conv3 | 63.60 | 60.96 | 58.64 | 81.81 |
| layer2.2.conv1 | 68.74 | 66.52 | 56.07 | 49.60 |
| layer2.2.conv2 | 70.42 | 69.81 | 69.10 | 87.02 |
| layer2.2.conv3 | 63.30 | 60.06 | 54.06 | 83.67 |
| layer2.3.conv1 | 63.87 | 59.87 | 44.49 | 43.37 |
| layer2.3.conv2 | 70.13 | 68.83 | 67.57 | 86.60 |
| layer2.3.conv3 | 66.09 | 62.03 | 49.41 | 82.74 |
| layer3.0.conv1 | 54.05 | 54.86 | 76.17 | 100.00 |
| layer3.0.conv2 | 82.47 | 89.52 | 99.25 | 100.00 |
| layer3.0.conv3 | 65.61 | 82.20 | 98.40 | 100.00 |
| layer3.0.downsample.0 | 78.21 | 71.87 | 54.37 | 53.19 |
| layer3.1.conv1 | 78.03 | 78.40 | 79.45 | 100.00 |
| layer3.1.conv2 | 82.93 | 92.45 | 98.20 | 100.00 |
| layer3.1.conv3 | 69.54 | 87.02 | 96.55 | 100.00 |
| layer3.2.conv1 | 74.19 | 72.72 | 89.19 | 100.00 |
| layer3.2.conv2 | 81.22 | 89.29 | 99.56 | 100.00 |
| layer3.2.conv3 | 68.24 | 82.87 | 99.03 | 100.00 |
| layer3.3.conv1 | 66.58 | 63.63 | 88.10 | 100.00 |
| layer3.3.conv2 | 78.70 | 88.28 | 99.54 | 100.00 |
| layer3.3.conv3 | 65.76 | 80.80 | 99.00 | 100.00 |
| layer3.4.conv1 | 57.57 | 59.99 | 75.83 | 100.00 |
| layer3.4.conv2 | 79.10 | 88.66 | 98.66 | 100.00 |
| layer3.4.conv3 | 62.33 | 80.67 | 97.62 | 100.00 |
| layer3.5.conv1 | 53.63 | 58.43 | 78.13 | 100.00 |
| layer3.5.conv2 | 79.08 | 88.88 | 99.04 | 100.00 |
| layer3.5.conv3 | 58.30 | 80.30 | 98.32 | 100.00 |
| layer4.0.conv1 | 90.52 | 100.00 | 100.00 | 100.00 |
| layer4.0.conv2 | 99.80 | 100.00 | 100.00 | 100.00 |
| layer4.0.conv3 | 99.37 | 100.00 | 100.00 | 100.00 |
| layer4.0.downsample.0 | 75.46 | 79.81 | 86.40 | 91.34 |
| layer4.1.conv1 | 97.11 | 100.00 | 100.00 | 100.00 |
| layer4.1.conv2 | 99.90 | 100.00 | 100.00 | 100.00 |
| layer4.1.conv3 | 99.56 | 100.00 | 100.00 | 100.00 |
| layer4.2.conv1 | 98.61 | 100.00 | 100.00 | 100.00 |
| layer4.2.conv2 | 99.96 | 100.00 | 100.00 | 100.00 |
| layer4.2.conv3 | 99.74 | 100.00 | 100.00 | 100.00 |
| fc | 83.95 | 82.61 | 81.69 | 86.05 |

Table 14: Sparsity budgets of SEW ResNet-18 using STDS + S-LATS on ImageNet.

| Final threshold $D$ | 0.5 | 1.0 | 2.0 | 3.0 | 5.0 | 7.0 | 10 | 15 | 20 |
|---|---|---|---|---|---|---|---|---|---|
| Top-1 Acc. (%) | 62.59 | 62.3 | 60.806 | 59.816 | 57.572 | 55.454 | 53.74 | 50.024 | 47.586 |
| Layer(s) | Sparsity (%) | | | | | | | | |
| Overall | 60.11 | 71.18 | 79.74 | 83.75 | 88.11 | 89.96 | 92.57 | 94.30 | 95.21 |
| conv1 | 38.70 | 49.83 | 62.40 | 66.90 | 73.93 | 79.44 | 80.71 | 85.43 | 86.90 |
| layer1.0.conv1.0 | 48.98 | 60.42 | 73.58 | 76.98 | 82.13 | 86.19 | 87.39 | 90.02 | 91.90 |
| layer1.0.conv2.0 | 37.88 | 50.73 | 61.17 | 66.69 | 71.79 | 76.28 | 79.39 | 82.53 | 84.40 |
| layer1.1.conv1.0 | 40.37 | 54.44 | 66.75 | 72.12 | 77.59 | 80.65 | 84.02 | 86.53 | 87.84 |
| layer1.1.conv2.0 | 39.94 | 51.82 | 62.50 | 67.47 | 73.18 | 77.47 | 80.40 | 84.16 | 85.53 |
| layer2.0.conv1.0 | 39.78 | 52.67 | 64.46 | 68.41 | 74.30 | 77.91 | 81.07 | 83.65 | 85.84 |
| layer2.0.conv2.0 | 46.68 | 59.70 | 70.55 | 74.26 | 79.64 | 83.32 | 85.99 | 88.92 | 90.62 |
| layer2.0.downsample.0.0 | 16.81 | 25.96 | 36.11 | 41.39 | 47.29 | 54.48 | 59.00 | 63.48 | 67.22 |
| layer2.1.conv1.0 | 48.87 | 62.95 | 73.58 | 77.19 | 81.24 | 84.64 | 87.51 | 89.59 | 91.89 |
| layer2.1.conv2.0 | 49.85 | 63.01 | 71.87 | 77.04 | 81.44 | 84.69 | 87.65 | 90.72 | 91.80 |
| layer3.0.conv1.0 | 48.98 | 60.98 | 71.23 | 75.30 | 80.16 | 83.48 | 85.91 | 88.49 | 90.02 |
| layer3.0.conv2.0 | 56.54 | 68.36 | 77.14 | 81.23 | 86.21 | 88.38 | 91.20 | 93.05 | 94.38 |
| layer3.0.downsample.0.0 | 27.11 | 37.56 | 48.76 | 54.21 | 61.76 | 66.29 | 70.50 | 75.62 | 79.10 |
| layer3.1.conv1.0 | 61.25 | 72.95 | 80.88 | 83.91 | 87.98 | 89.96 | 91.59 | 93.66 | 94.48 |
| layer3.1.conv2.0 | 60.16 | 71.00 | 79.03 | 82.63 | 86.38 | 88.59 | 90.49 | 92.50 | 93.61 |
| layer4.0.conv1.0 | 58.92 | 70.02 | 78.17 | 82.08 | 86.23 | 88.38 | 90.44 | 92.65 | 93.86 |
| layer4.0.conv2.0 | 64.63 | 74.90 | 82.40 | 85.93 | 89.74 | 91.58 | 93.27 | 95.14 | 95.96 |
| layer4.0.downsample.0.0 | 31.91 | 42.72 | 53.25 | 58.97 | 65.57 | 69.29 | 72.34 | 75.55 | 77.45 |
| layer4.1.conv1.0 | 67.33 | 77.95 | 85.81 | 89.57 | 93.42 | 95.61 | 96.99 | 97.89 | 98.38 |
| layer4.1.conv2.0 | 63.04 | 72.57 | 80.14 | 84.08 | 88.71 | 91.79 | 94.04 | 95.35 | 96.01 |
| fc | 33.00 | 52.40 | 71.29 | 79.66 | 86.71 | 89.96 | 92.52 | 94.95 | 96.27 |

Table 15: Sparsity budgets of SEW ResNet-18 using our implementation of original STDS (STDS + Sine scheduler) on ImageNet.

| Final threshold $D$ | 0.6 | 0.8 | 1.5 | 3.0 | 5.0 |
|---|---|---|---|---|---|
| Top-1 Acc. (%) | 61.114 | 60.218 | 57.458 | 52.966 | 48.436 |
| Layer(s) | Sparsity (%) | | | | |
| Overall | 76.06 | 79.91 | 85.91 | 90.62 | 93.19 |
| conv1 | 48.07 | 53.99 | 68.15 | 79.49 | 85.71 |
| layer1.0.conv1.0 | 61.37 | 69.80 | 81.65 | 88.07 | 91.37 |
| layer1.0.conv2.0 | 56.20 | 62.73 | 73.62 | 79.45 | 83.24 |
| layer1.1.conv1.0 | 56.73 | 64.55 | 79.27 | 84.19 | 87.55 |
| layer1.1.conv2.0 | 58.62 | 65.86 | 73.35 | 79.39 | 84.28 |
| layer2.0.conv1.0 | 58.34 | 66.05 | 75.50 | 80.44 | 85.06 |
| layer2.0.conv2.0 | 66.05 | 72.17 | 80.13 | 85.38 | 88.55 |
| layer2.0.downsample.0.0 | 28.70 | 36.28 | 49.60 | 58.87 | 64.78 |
| layer2.1.conv1.0 | 68.32 | 74.54 | 82.47 | 87.28 | 89.96 |
| layer2.1.conv2.0 | 68.65 | 73.85 | 80.92 | 86.52 | 89.55 |
| layer3.0.conv1.0 | 68.04 | 73.05 | 80.11 | 86.17 | 88.89 |
| layer3.0.conv2.0 | 74.94 | 79.00 | 85.20 | 89.69 | 92.33 |
| layer3.0.downsample.0.0 | 44.28 | 50.82 | 62.14 | 70.68 | 76.31 |
| layer3.1.conv1.0 | 78.62 | 82.50 | 87.89 | 91.36 | 93.57 |
| layer3.1.conv2.0 | 76.93 | 80.73 | 86.32 | 90.13 | 92.28 |
| layer4.0.conv1.0 | 76.41 | 80.41 | 86.30 | 90.59 | 92.95 |
| layer4.0.conv2.0 | 80.60 | 83.84 | 88.43 | 92.07 | 93.92 |
| layer4.0.downsample.0.0 | 49.57 | 55.31 | 63.96 | 71.63 | 75.94 |
| layer4.1.conv1.0 | 82.60 | 85.81 | 90.74 | 94.64 | 96.62 |
| layer4.1.conv2.0 | 77.17 | 79.68 | 84.09 | 88.96 | 92.32 |
| fc | 46.12 | 56.43 | 77.07 | 90.62 | 95.18 |

