# OpenReview forum: "A Unified Framework for Soft Threshold Pruning"
_ICLR.cc/2023/Conference — ICLR 2023 poster_

### Official Review · Reviewer_GHaU · 2022-10-23

**Confidence:** 3
**Correctness:** 4
**Technical Novelty And Significance:** 3
**Empirical Novelty And Significance:** 3
**Recommendation:** 6

**Clarity, Quality, Novelty And Reproducibility:**

This paper is clearly written and the quality is good. The proposed method is valid and the novelty of this paper is good. The experiments in this paper are reproducible.

**Details Of Ethics Concerns:**

There are no ethics concerns.

**Strength And Weaknesses:**

Strength:
1. The paper is well written and the presentation is clear.
2. The pruning problem is an important one.
3. The theoretical soundness is good.

Weaknesses:
1. The method is evaluated on relatively a small number of networks. Whether the performance improvement persists for other network architectures, such as recurrent neural networks?
2. How the performance is influenced by hyper-parameters? Whether the performance improvement is robust under varying hyper-parameters?

**Summary Of The Paper:**

This paper has reformulated soft threshold pruning as an implicit optimization problem solved using the Iterative Shrinkage-Thresholding Algorithm (ISTA), and proposed a unified theoretical framework for soft threshold pruning. Experiments have shown that the proposed method outperforms SOTA soft threshold pruning algorithms, such as STR and STDS.

**Summary Of The Review:**

The considered problem is very important. The proposed framework is technically sound and achieves better performance than existing SOTA methods. It would be nice to test the method on more architectures and diverse hyper-parameter settings.

---

> ### Author Response · Authors · 2022-11-16
> **Author rebuttal**
>
> To reviewer GHaU, we thank you for your viewpoints on our manuscript! We are delighted to address your concerns.
>
> - The method is evaluated on relatively a small number of networks. Whether the performance improvement persists for other network architectures, such as recurrent neural networks?
>
> As shown in the text, our evaluation includes ANNs and SNNs. The SNNs have time-dependent hidden variables and thus can be viewed as RNNs (See Appendix G.1 for detail).
>
> Alternatively, the question may refer to other kinds of conventional RNNs like LSTM. We will add further experiments on those RNNs in future revisions. We would welcome any suggestions on which representative architectures in RNNs should serve as testbeds.
>
> - How the performance is influenced by hyper-parameters? Whether the performance improvement is robust under varying hyper-parameters?
>
> Generally speaking, changes to training hyper-parameters will influence both the accuracy and sparsity in sparsify-during-training methods. Comparison between different sets of training hyper-parameters tends to be viewed as unfair for they may lead to differences in dense baselines. For this reason, most work adheres to the same set of training hyper-parameters (this work, WoodFisher, STR, etc.) and we thereby don't change them. Nonetheless, we still try increasing batch size since some recent works adopt such a setting and achieve higher accuracy, as stated in section 6.1, which indicates the improvement is robust across different batch sizes.
>
> The question may also be about individual hyper-parameters in pruning. We explain at the end of the first paragraph in section 6 that the only tunable hyper-parameter in our work is the final threshold $D$ when the scheduler function is given. *Increasing $D$ alone* achieves higher sparsity (equivalent to increasing L1 regularization) without touching any other hyperparameters. We also rerun our experiments as requested by reviewer 3EDJ to show that the performance improvement is solid.

---

### Official Review · Reviewer_Lscv · 2022-10-26

**Confidence:** 3
**Correctness:** 4
**Technical Novelty And Significance:** 3
**Empirical Novelty And Significance:** 3
**Recommendation:** 6

**Clarity, Quality, Novelty And Reproducibility:**

- Clarity: The paper is cleanly written in the general. Some issues are listed in detailed comments.

- Quality: The paper is technically sound.

- Novelty: As mentioned above, the framework with local ISTA is not really new for network pruning. More innovations come from the theory grounded pruning threshold.

- Reproducibility: Code is provided for reproduction.





**Strength And Weaknesses:**

Strengths:

- The paper establishes nice connections between the learning rate and pruning threshold scheduler via Theorem 1.

- The methodologies developed in this paper are well grounded in theory, e.g., the learning rate adapted optimal threshold scheduler for pruning, and PGH schedulers that connects to early pruning.

- The authors conduct extensive studies across various network architectures (ANNs & SNNs) as well as different pruning settings.

Weakness:

- While I appreciate the authors' effort in introducing the optimal threshold scheduler and sibling schedulers for pruning, it is not new to formulate pruning with local ISTA, which has been studied by plenty of previous works, e.g., [1,2,3].

  - [1] He Y, Zhang X, Sun J. Channel pruning for accelerating very deep neural networks. Proceedings of the IEEE international conference on computer vision. 2017: 1389-1397.

  - [2] Zhang D, Wang H, Figueiredo M, et al. Learning to share: Simultaneous parameter tying and sparsification in deep learning, International Conference on Learning Representations. 2018.

  - [3] Bai H, Wu J, King I, et al. Few shot network compression via cross distillation. Proceedings of the AAAI Conference on Artificial Intelligence. 2020, 34(04): 3203-3210.

- Only unstructured pruning is discussed in the paper, this however, has limited acceleration in practice. I wonder whether similar results can be established for structured pruning, e.g., via $\mathcal{L}_{2,1}$ regularization.

Detailed comments:

- Some parts of the paper are not self-contaiend. While it is claimed to be verified on Mobilenet-V1 and spiking neural networks, their results are in the appendix and not referenced in the main text.

- Section 3.1 could have been organized better. There is some notation overlapping (e.g., $d$, $\phi(\cdot)$ and $g(\cdot)$ for the pruning threshold). It would be better to have a short overview of STR and STDS first.

- Provide more details on how Eq. 12 and Eq. 13 are derived.


**Summary Of The Paper:**

This paper presents a unified framework of iterative soft threshold pruning (ISTA). The framework reveals that previous studies with threshold pruning can be casted as adding regularization terms (i.e., L1-regularization) for training. The authors further connect the learning rate with pruning threshold, and propose the learning rate adapted optimal threshold scheduler. The framework also has the link to other pruning methods such as early pruning, pruning at initialization. Experimental results on various network architectures (ResNet-50, MobileNet-V1 and Spiking Neural Networks) demonstrate the effectiveness of the approach.

**Summary Of The Review:**

The paper provides a comprehensive and theoretical study of soft threshold pruning. While it is not new to tackle network pruning with ISTA or proximal mapping of $\mathcal{L}_1$ regularizer, the theory developed in this paper further motivates the design of optimal pruning threshold scheduler. The experiments are also adequate and effective to demonstrate the proposed solutions. The authors also provide in-depth analysis on how the framework relates early pruning, spiking neural nets, and more in the appendix. One potential issue with the paper is the practicality, as unstructured pruning can hardly bring actual acceleration. Overall, the paper is quite informative and well presented.

---

> ### Author Response · Authors · 2022-11-16
> **Author rebuttal**
>
> To reviewer Lscv, we thank you for your helpful feedback and appreciation of the theory grounded-pruning threshold! We are willing to address all the concerns.
>
> - While I appreciate the authors' effort in introducing the optimal threshold scheduler and sibling schedulers for pruning, it is not new to formulate pruning with local ISTA, which has been studied by plenty of previous works
>
> Compared to mentioned works, our paper takes the initiative to construct a unified framework based on ISTA, to which popular soft threshold pruning methods could fit. We have cited the aforementioned studies in the revision.
>
> We would like to point out that we are not the first to use ISTA for network pruning, and we also do not overclaim it in the text. However, this work is the first to unify a series of existing pruning algorithms using ISTA. It discovers the underlying implicit ISTA in soft threshold pruning, introduces a unified framework, and improves the performance based on the above understandings. It allows us to compare and contrast different methods in a fair manner, which also enable us to view schedulers from a higher level, and thus to revisit the common beliefs and design decisions of the existing approaches.
>
> We will add a clear comparison between our method and the mentioned works in future versions.
>
> - Only unstructured pruning is discussed in the paper, this however, has limited acceleration in practice. I wonder whether similar results can be established for structured pruning, e.g., via $\mathcal{L}_{2,1}$ regularization.
>
> Thanks for the kind suggestions! $L_{2,1}$ regularization is a widespread technique in structured pruning, which is also well-validated in [1] and [2]. However, it is not compatible with soft threshold reparameterization. In fact, in soft threshold pruning, $L_{2,1}$ regularization does not correspond to an update rule or reparameterization function. This is because the inner $L_2$ norm of $L_{2,1}$ regularization is inseparable and hence the solution of Eq.4 cannot derive an individual update rule for each component $w_i$ in network weight $\boldsymbol{w}$.
>
> Fortunately, $L_{2,1}$ regularization is not the only way to induce structured sparsity. [3] proposes a simple but effective idea that the learnable scale parameter $\gamma$ in the affine transformation $z_{out}=\gamma\hat{z}+\beta$ of BatchNorm layers can be used for channel pruning. In this way, the unstructured pruning to $\gamma$ turns into structured (channel) pruning. Based on [3], we change the affine transformation a little to $z_{out}=\gamma(\hat{z}+\beta)$ to ensure both weight and bias are pruned. After that, the only thing we need is to reparameterize $\gamma$ using soft threshold mapping and prune it with our method. We will report the accuracy and sparsity in future revisions.
>
>
>
> [1] Wen W, Wu C, Wang Y, Chen Y, Li H. Learning structured sparsity in deep neural networks. Advances in neural information processing systems. 2016;29
>
> [2] Zhou H, Alvarez JM, Porikli F. Less is more: Towards compact cnns. In European conference on computer vision. 2016 (pp. 662-677).
>
> [3] Liu Z, Li J, Shen Z, Huang G, Yan S, Zhang C. Learning efficient convolutional networks through network slimming. In Proceedings of the IEEE international conference on computer vision. 2017 (pp. 2736-2744).
>
> - Some parts of the paper are not self-contaiend. While it is claimed to be verified on Mobilenet-V1 and spiking neural networks, their results are in the appendix and not referenced in the main text.
>
> We actually did reference them in section 6.1 of the submitted version. We will also ensure all contents in Appendix are referenced in the main text in future revisions. As can be seen, we are short of space for organizing all results (including those further requested by the reviewers) into the main text. We do our utmost to balance the spaces between each part and ultimately find that there must be some results left in the appendix.
>
> - Section 3.1 could have been organized better. There is some notation overlapping (e.g., $d$, $\phi(\cdot)$ and $g(\cdot)$ for the pruning threshold). It would be better to have a short overview of STR and STDS first.
>
> Thanks for the constructive comment! We reorganize section 3.1 in the revision and also lighten the notations in Algorithm 1 with greater clarity. We first introduce the core steps in soft threshold pruning as an overview and then expound upon the differences between methods.
>
> - Provide more details on how Eq. 12 and Eq. 13 are derived.
>
> The detailed derivation of Eq.12 and Eq.13 is provided in Appendix D of the revision. We reference them in the last paragraph of section 5.1.

---

### Official Review · Reviewer_3EDJ · 2022-11-04

**Confidence:** 3
**Correctness:** 2
**Technical Novelty And Significance:** 3
**Empirical Novelty And Significance:** 3
**Recommendation:** 3

**Clarity, Quality, Novelty And Reproducibility:**

The clarity and quality of this paper is quite low for the reasons outlined above in the weakness section. I will list a few more instances where the clarity and quality needs significant work below. The paper is reasonably novel. To the best of my knowledge, such a theoretically grounded analysis of threshold scheduler constitutes original work. The paper is not reproducible for that important methodological details are missing. See my first point under Weakness #2.

Here are a non-exhaustive list of instances in the paper where clarify and quality is clearly lacking:
- On page 1, what do the authors mean by this and what purpose does this sentence serve: "vanilla magnitude-based pruning is considered only as a baseline method."
- On page 2, when the authors describe the merits of ISTA algorithm, the authors wrote "[ISTA] has long been certified as an effective 'pruning' algorithm in all sorts of fields". The author provides neither details on what field is ISTA algorithm effective within nor any citations backing up this claim.
- On page 2, what do the authors mean by "the rationality of training threshold"?
- On page 2, what do the authors mean by "stable regularized term implies a coherent pace of the foregoing two schedulers"? I do not understand this even after reading the whole paper.
- On contribution #3, why should people care about early soft threshold pruning algorithm? From my understanding, soft threshold pruning algorithm does not improve training efficiency because the pruned weights are not actually removed. Then what is the point of pruning early when applying soft threshold pruning?
- On page 3, Sec 3.1, what is n?
- Again on page 3, Sec 3.1, GPO is only defined as one of 12 related works on sparsify-during-training that the authors cited (without any description/explaination). The authors should not expect this term to still exist in the working memory of the audience.
- On page 4, in Algorithm 1, what is sigma? It is never defined.

**Strength And Weaknesses:**

Strength:
- The authors aspire to come up with a unified theoretical backbone of a variety of soft thresholding pruning algorithm. This is an important problem because the community can benefit from a shared theoretical infrastructure for analyzing threshold pruning techniques, which are often empirical in nature.
- The authors propose a theoretically grounded threshold scheduler and demonstrate  some benefits of this threshold scheduler empirically.

Weakness:
- The paper is highly disorganized on at least two levels.
    - **Inadequate explanation of core concepts.** The authors made little to no attempt to introduce and define essential concepts such as "threshold scheduler" for the audience early in the paper. As a result I have to start with Sec.3 Preliminaries and work my way back to make sense of the abstract and introduction.
    - **Disorganized prsentation of technical details.** This disorganized style of writing manifests even when the author discusses specific technical details. For example, when the authors introduce Eq.2, the symbol \mu appears out of nowhere. Upon further examination, I realize that this variable \mu is implicitly defined in the L.3 of the pseudocode presented in Algorithm. 1 as the output of the learning rate scheduler function. The author must clearly define symbols before using them.

        While the authors may very well have much to share with and educate their audience, the disorganized writing style is a significant obstacle to achieving this potential.

- There are major red flags in experimental evaluation.
	- **Reproducibility and fairness of comparison**. I believe authors obtain individual data points in Fig. 2 comparing the proposed threshold scheduler with alternatives by "tuning final threshold D". But the author does not provide details for how this threshold is tuned. This raises questions on 1). whether the core empirical evaluation of this paper can be reproduced 2). whether comparison with alternative soft threshold pruning techniques is fair.
	- **Missing simple baseline.** For the left plot of Fig. 2, the authors instrumented a prior soft threshold pruning technique called STDS to use their proposed threshold scheduler and demonstrated superior performance to other soft threshold pruning techniques. However, it is unclear whether this superior performance is due to the STDS pruning algorithm or the proposed threshold scheduler. The author should report performance of STDS with the default threshold scheduler as an important baseline.
	- **No characterization of uncertainties.** The performance of pruning algorithms can have significant variance between different weight initialization/SGD data order. The results presented in this work lack characterization of uncertainties stemming from this variance.

**Summary Of The Paper:**

This paper drew a connection between soft threshold pruning and Iterative Shrinkage-Thresholding Algorithm (ISTA). Based on this interpretation of soft threshold pruning the authors derive novel threshold schedulers that appear to perform better than alternatives in terms of sparsity-accuracy trade-off.

**Summary Of The Review:**

While the authors investigated an important problem and conducted original research, the quality of presentation and empirical evaluation is significantly below that required for acceptance into ICLR.

---

> ### Author Response · Authors · 2022-11-15
> **Author rebuttal**
>
> **Lacking in clarity and quality**
>
> - On page 1, what do the authors mean by this and what purpose does this sentence serve: "vanilla magnitude-based pruning is considered only as a baseline method."
>
> We have removed this sentence since it is indeed irrelevant to our point.
>
> - On page 2, when the authors describe the merits of ISTA algorithm, the authors wrote "[ISTA] has long been certified as an effective 'pruning' algorithm in all sorts of fields". The author provides neither details on what field is ISTA algorithm effective within nor any citations backing up this claim.
>
> We rephrase it by replacing "'pruning' algorithm" with "sparsification method". We also provide a bunch of supporting citations on the fields of application of ISTA for this statement, including deep learning, computer vision, medical imageology and geophysics.
>
> - On page 2, what do the authors mean by "the rationality of training threshold"?
>
> We actually explain why we should not train threshold as STR or GPO did in Appendix C. However, there is little space for placing them in main texts. We are still discussing where we should emphasize this point in the main text.
>
> - On page 2, what do the authors mean by "stable regularized term implies a coherent pace of the foregoing two schedulers"? I do not understand this even after reading the whole paper.
>
> It refers to Corollary 1 that the stable L1 regularized term $\mu\\|\boldsymbol{w}\\|_1$ requires the update of threshold $d^{(t+1)}-d^{(t)}$ to be proportional to learning rate $\eta^{(t)}$ with coefficient $\mu$. We also highlight it in the first paragraph of section 5.1 in revision.
>
> - On contribution #3, why should people care about early soft threshold pruning algorithm? From my understanding, soft threshold pruning algorithm does not improve training efficiency because the pruned weights are not actually removed. Then what is the point of pruning early when applying soft threshold pruning?
>
> We respectfully disagree with the statement that early soft threshold pruning does not improve training efficiency. The pruning mask can be fixed when the sparsity is no longer increasing as shown in the left of Fig.3. We will add experiments on fixed connectivity pattern using PGH scheduler as an early pruning algorithm.
>
> - On page 3, Sec 3.1, what is n?
>
> We rephrase this without using $n$ since it is otherwise defined in section 5.1.
>
> - Again on page 3, Sec 3.1, GPO is only defined as one of 12 related works on sparsify-during-training that the authors cited (without any description/explaination). The authors should not expect this term to still exist in the working memory of the audience.
>
> We add a brief declaration in section 3.1 that the GPO methods will fall back to STR and shares the same discussion with STR. The analysis is now given in Appendix C.2.
>
> - On page 4, in Algorithm 1, what is sigma? It is never defined.
>
> It refers to sigmoid function, we now relocate the definition to the beginning of section 3.

---

> > ### Comment · Reviewer_3EDJ · 2022-11-19
> > **Response**
> >
> > I acknowledge that the author's response has addressed most of my concerns around the experimental evaluation protocol. Though not required, I recommend briefly describing the procedure with which you increased the final threshold D to reach certain sparsities because some of the final values of D look quite arbitrary and it is unclear to me how you obtained them.
> >
> > In terms of paper presentation, the updated draft is an improvement over the previous one, however it is still far from the quality required for acceptance to ICLR. My largest concern is that I cannot understand a significant fraction of the paper as is, and for this reason I doubt this paper will be valuable to anyone outside a handful of people who already possess the required background knowledge for ISTA and to a lesser extent, threshold pruning. The author response and the newly introduced edits do help improve the quality of the presentation. However, multiple rounds of feedback and edits are required to get this paper to an acceptable quality, which is not possible during this brief reviewing window. That said, I will detail several obvious things to improve in the future at the end.
> >
> > Thus, I stand by my assessment to recommend rejection.
> >
> > Some detailed feedback, please do not interpret this list as my rationale for rejection:
> >
> > -  You need to introduce the core concepts associated with soft threshold pruning in the introduction. Without such an introduction, the first paragraph on P2 beginning with "In the past few years, " makes no sense to anyone who is not familiar with soft threshold pruning and its associated terminologies already. For example, what does "soft threshold reparameterization" mean for a reader who is not already familiar with soft threshold pruning?
> >
> > - You response does not clarify my confusion with the phrase "stable regularized term implies a coherent pace of the foregoing two schedulers" used to describe your contribution. I do not understand what "stable", "coherent pace", "foregoing" mean in the context of your work and I would recommend rephrase with more concrete terms.
> >
> > - On P1 line 4, embedding devices -> embedded device.
> >
> > - On P1 second paragraph, when you say pruning is an optimization problem under L0 norm constraint, you are talking about a specific pruning setup, of which there are many in the existing literature. What is the objective being optimized for, is it accuracy? If so I would say it explicitly. Another setup for pruning is to minimize the number of parameters while holding accuracy constant, I think you can more clearly disambiguate from this arguably more common setup.
> >
> > - On P2 what do you mean by "orthonormal design case"?
> >
> > - On P5 line 2, I think you mean "leave the regularization alone"?
> >
> > - Again on this page, what kind of expansion of F yields (2)? Assuming this is Taylor expansion, I don't see where \eta comes from.

---

> > > ### Author Response · Authors · 2022-11-25
> > > **Some clarifications on technical details**
> > >
> > > We would like to thank you for your kind suggestions for the clarity issues, which for sure will make our work easier to understand for a wider variety of readers. We still try to explain some vital technical facts that we may not make very clear.
> > >
> > > > On P1 second paragraph, when you say pruning is an optimization problem under L0 norm constraint, you are talking about a specific pruning setup, of which there are many in the existing literature. What is the objective being optimized for, is it accuracy? If so I would say it explicitly. Another setup for pruning is to minimize the number of parameters while holding accuracy constant, I think you can more clearly disambiguate from this arguably more common setup.
> > >
> > > To our knowledge, the objective of pruning is clear, which is higher performance (lower loss) with L0 norm constraint (hard or soft). A pruning study could fix one metric to optimize the other, like using fewer parameters to achieve the same accuracy or to achieve higher performance under the same sparsity. A hard L0 constraint like $\min_{w}\mathcal{L}(w),\\|w\\|_0\leq c$ corresponds to the setting *higher performance under the same sparsity*. Adding the L0 norm as an explicit regularized term and relaxing it to the tightest convex L1 norm as ISTA do is a "soft" version of the above problem. Meanwhile, although rare, a pruning study can certainly choose to optimize the L0 norm while maintaining loss and it becomes a combinatorial optimization problem. However, the two formulations all fall into an optimization problem with the L0 norm constraint.
> > >
> > > Last but not least, the overall performance of a pruning algorithm can always be evaluated and compared using a sparsity vs accuracy curve as shown in Fig.2 no matter what specific formulation of pruning it follows. *Different formulations of pruning as optimization problems do not affect the evaluation of the performance of pruning algorithms.*
> > >
> > > > On P2 what do you mean by "orthonormal design case"?
> > >
> > > It refers to the case when the design matrix in LASSO is orthonormal (See section 10 in LASSO [1]).
> > >
> > > > Again on this page, what kind of expansion of F yields (2)? Assuming this is Taylor expansion, I don't see where \eta comes from
> > >
> > > It is a common expression in optimization literature and you may find it in optimization literature like FISTA [2]. Strictly speaking, the Taylor expansions have the form
> > > $$
> > > \hat{F}_{\eta}(\boldsymbol{y};\boldsymbol{x}):=f(\boldsymbol{x})+\langle\boldsymbol{y}-\boldsymbol{x},\nabla f(\boldsymbol{x})\rangle+\frac{1}{2}(\boldsymbol{y}-\boldsymbol{x})^\top\nabla^2f(\bar{\boldsymbol{y}})(\boldsymbol{y}-\boldsymbol{x})+r(\boldsymbol{y})
> > > $$
> > > where $\bar{\boldsymbol{y}}$ lies on the line segment connecting $\boldsymbol{x}$ and $\boldsymbol{y}$. Since $f$ is $L$-smooth, we have $\nabla^2 f\\preccurlyeq LI$. Then the second-order term is bounded between $-\frac{L}{2}\\|\boldsymbol{y}-\boldsymbol{x}\\|_2^2$ and $\frac{L}{2}\\|\boldsymbol{y}-\boldsymbol{x}\\|_2^2$ and can thus be expressed in form of $\frac{1}{2\eta}\\|\boldsymbol{y}-\boldsymbol{x}\\|_2^2$ with $|\eta|\geq1/L$. If $f$ is convex, we can say $\eta>0$ and get a positive stepsize for $\nabla^2 f\\succcurlyeq 0$. When $f$ is non-convex, $\eta$ can be negative but we still assume a positive stepsize to optimize the upperbound.
> > >
> > > [1] Tibshirani R. Regression shrinkage and selection via the lasso. Journal of the Royal Statistical Society: Series B (Methodological). 1996;58(1):267-88.
> > >
> > > [2] Beck A, Teboulle M. A fast iterative shrinkage-thresholding algorithm for linear inverse problems. SIAM journal on imaging sciences. 2009;2(1):183-202.

---

> > > > ### Comment · Reviewer_3EDJ · 2022-11-25
> > > > **Reponse**
> > > >
> > > > > To our knowledge, the objective of pruning is clear, which is higher performance (lower loss) with L0 norm constraint (hard or soft).
> > > >
> > > > > Meanwhile, although rare, a pruning study can certainly choose to optimize the L0 norm while maintaining loss and it becomes a combinatorial optimization problem.
> > > >
> > > > I disagree the pruning community has a consensus on the objective of pruning. While your setup (lower loss with L0 constraint) is certainly reasonable, an influential line of work exists that explores this other setup you mentioned (optimize the L0 norm while maintaining loss), including [1] and [2].
> > > >
> > > > > However, the two formulations all fall into an optimization problem with the L0 norm constraint.
> > > >
> > > > It is unacceptable to address ambiguity in your text by claiming that both interpretations can be valid.
> > > >
> > > > Two of your remaining responses corroborate one of my central concerns -- this paper is not self-contained. I do not have the background in optimization literature to make sense of the core result you present in the main body of the paper. As a reader eager to make sense of your work, I even hunted down the only paper you referenced (Martinet, 1970) in this section to gather the requisite context only to find it written in French.
> > > >
> > > > [1] Learning both Weights and Connections for Efficient Neural Networks
> > > >
> > > > [2] The Lottery Ticket Hypothesis: Finding Sparse, Trainable Neural Networks

---

> ### Author Response · Authors · 2022-11-15
> **Author rebuttal**
>
> To reviewer 3EDJ, we thank you for the careful reading and insightful comments! We endeavor to resolve all your concerns.
>
> **Inadequate explanation of core concepts**
>
> We rephrase the section 3.1 and highlight some core concepts and terminologies such as *threshold scheduler* and *scheduler function* used in this paper. Hopefully, this could lead to less confusion for readers.
>
> **Disorganized presentation of technical details**
>
> Thanks for your careful reading! We check throughout the text and reorganize the definition of notations to ensure they are always defined in advance. However, we find there is no $\mu$ in Line 3 in pseudocode of Algorithm 1 in the submitted version, which is $\eta^{(t)}\leftarrow\varphi(t,T)$. We venture to speculate your concern is about definition of $\eta$. Hence, we add it in section 3.2 that the $\eta$ in Eq.2 is explained as learning rate or stepsize. Note that some notations are moved to the beginning part of section 3 to avoid confusions.
>
> **Reproducibility and fairness of comparison**
>
> - Details on "tuning final threshold D"
>
> We add further clarification on both the "Threshold tuning" part in section 3.1 and the end of the first paragraph in section 6, respectively, to explain that we *increase D* to achieve higher sparsity (equivalent to increasing L1 regularization) without touching any other hyperparameters. Since none of training hyperparameters is changed, we believe this leads to a fair comparison with previous work.
>
> - Whether can be reproduced
>
> Since the only changing hyperparameter is D, we have reported D presented in all experiments for the time being in Appendix I. It should be easy to reproduce using codes provided in supplementary.
>
> **Missing simple baseline**
>
> Thanks for the constructive comment. To evaluate the mentioned baseline both on ANNs, we add a formal ablation study in Appendix F, which is also referred to in section 6.1. As shown in Fig.6, our proposed threshold scheduler clearly surpasses the default schedulers of STDS by a large margin.
>
> **No characterization of uncertainties**
>
> We are running all previous trials for another 2 times without fixed random seeds to evaluate the uncertainties over 3 trials. Now we have replaced the performances on ResNet-50 of 256 batch size with ones with standard deviations. It shows the performance variation is not high enough to affect the conclusion. Rerunning all presented trials in our texts takes time, but we are working on it.

---

### Author Response · Authors · 2022-11-25
**Structured pruning**

To all reviewers, we would like to share some new results on the structured pruning setting. As discussed in our response to reviewer Lscv, we use a reparameterized BatchNorm (BN for short) layer to induce channel pruning. It is achieved by modifying the affine transformation $z\_{out}=\\gamma\\hat{z}+\\beta$ to $z\_{out}=\\gamma(\\hat{z}+\\beta)$ and prune $\\gamma$ using soft threshold reparameterization $\\gamma=\\mathcal{S}\_{d}(\\theta_{\\gamma})$ to sparsify both weight and bias of BN layer. If a layer lies between two BN layers with sparsity $p_{pre}$ and $p_{post}$, respectively, the sparsity of the intermediate layer is calculated by $1-(1-p\_{pre})(1-p\_{post})$. The FLOPs thereby decreases by $1-(1-p_{pre})(1-p_{post})$. The FLOPs of the first and the last layer are evaluated by the sparsity of the only nearby BN layers.

One of the current SOTA studies is ResRep [1], which trains ResNet-50 for 180 epochs on ImageNet datasets. We run our structured pruning version on the same setting and report our results with the other SOTA works as follows

| Method       | Pruned Top-1   | GFLOPs | FLOPs$\downarrow$ (%) |
| ------------ | -------------- | ------ | --------------------- |
| ResRep [1]   | $76.15\pm0.01$ | 1.8589 | 54.54                 |
| ResRep [1]   | $75.97\pm0.02$ | 1.7947 | 56.11                 |
| ResRep [1]   | $75.30\pm0.01$ | 1.5498 | 62.10                 |
| EagleEye [2] | $77.1$         | 3.0    | 26.64                 |
| EagleEye [2] | $76.4$         | 2.0    | 51.09                 |
| EagleEye [2] | $74.2$         | 1.0    | 75.54                 |
| **Ours**         | $76.33\pm0.10$ | 1.8794 | 54.04                 |
| **Ours**         | $75.62\pm0.07$ | 1.5691 | 61.63                 |

It is evident that with the simple idea of pruning BN, our method achieves **almost SOTA performances** on structured pruning. Note that combining our method with a forerunner work [3] is rather trivial and the performance may be further improved with advanced methods in structured pruning.

[1] Ding X, Hao T, Tan J, Liu J, Han J, Guo Y, Ding G. ResRep: Lossless CNN Pruning via Decoupling Remembering and Forgetting. In Proceedings of the IEEE/CVF International Conference on Computer Vision 2021 (pp. 4510-4520).

[2] Li B, Wu B, Su J, Wang G. EagleEye: Fast Sub-net Evaluation for Efficient Neural Network Pruning. In European conference on computer vision 2020 (pp. 639-654).

[3] Liu Z, Li J, Shen Z, Huang G, Yan S, Zhang C. Learning efficient convolutional networks through network slimming. In Proceedings of the IEEE international conference on computer vision. 2017 (pp. 2736-2744).

---

### Decision · Program_Chairs · 2023-01-20

**Decision:**

Accept: poster

**Justification For Why Not Higher Score:**

This paper is a nice contribution to a niche corner (soft threshold pruning) of a niche corner (unstructured neural network pruning) of a somewhat niche corner (neural network pruning) of the neural network literature. It tidies up this corner (which is great), but it is not of sufficiently broad interest to deserve a spotlight.

**Justification For Why Not Lower Score:**

The paper makes a solid technical advance in a relatively popular space.

**Metareview: Summary, Strengths And Weaknesses:**

Summary: This paper unifies the literature on various pruning methods that use soft thresholding. (Yes, this is a rehash of the paper's title, but the paper's title is very descriptive.) These pruning methods learn a score parameter alongside each weight, and only the weights with the highest sigmoid(score) parameter are kept. The authors generalize this framework in a way that subsumes several prior papers, provide theoretical justification for how to set the many hyperparameters of this class of methods (scheduling the threshold and regularizing the score parameters), and produce a pruning algorithm that appears to outperform prior work on standard benchmarks.

Strengths: This paper solves the technical problem convincingly.

Weaknesses:
* I agree with reviewer 3EDj that this paper is quite obtuse writing-wise in many places. It's not easy to parse unless you're familiar with both the pruning literature and the optimization literature. This paper is not self-contained and, even if the technical results are nice, the authors are limiting the potential influence of this work by requiring so much prior knowledge and skipping around to make sense of the paper (I read it and had the same challenges as reviewer 3EDj).
* This paper implicitly has a strong degree of "theoretical inevitable-ism" - the idea that, because formal math says something should work according to a simplified set of assumptions, it will work in practice in large-scale settings. That isn't true in much of deep learning (e.g., classical bias-variance tradeoff vs. double descent, NTK results having limited practical applicability), and I wish the authors were more careful about acknowledging that the derivations are a source of inspiration for good empirical results that are not inevitable merely because the authors did some proofs. This is a problem throughout the pruning literature, and the authors should not be held solely responsible for this, but they do have an opportunity to do better.

Overall: This is a probable accept. The technical results are solid, it brings order to a chaotic area of the literature, and the evaluation shows that the techniques in this paper improve the state-of-the-art in the process. The concerns of Reviewer 3EDj are acknowledged, and I hope the authors clean up this paper and make it accessible to a wider audience. Otherwise, in its current state, it isn't sufficiently accessible to the pruning community to have impact, and I worry it will collect dust as a contribution that isn't built on because it is too obtuse.

**Note From Pc:**

if the above contains the word "oral" or "spotlight" please see: "oral" presentation means -> notable-top-5% and "spotlight" means -> notable-top-25%. As stated in our emails, we are disassociating presentation type from AC recommendations

**Summary Of Ac-Reviewer Meeting:**

N/A